# Criterion-Conditional In-Context Learning:
# Evaluating Criterion-Shift Adaptation in Vision-Language Models

**Kaiyun Yang** [* 1]   **Ruilin Yang** [* 2]   **Zhimin Yao** [2]   **Jikai Wang** [1]   **Wei Ge** [2]

## Abstract

Vision-language models can perform new tasks without parameter updates through in-context learning (ICL), whose core mechanism is utilizing the support set for task induction. In the standard ICL setting, once the task is induced, its decision criterion remains fixed. However, in real-world applications, many tasks exhibit a stable high-level intent, while their decision criteria shift according to specific requirements. Thus, we introduce a new setting, denoted as Criterion-Conditional In-Context Learning (CC-ICL), where models must infer the latent criterion from context and adjust predictions accordingly under fixed task semantics. To evaluate this capability, we propose two complementary metrics, Criterion Invariance and Criterion Sensitivity, capturing the model's robustness and adaptability under criterion shifts. We further construct CC-Bench, a multi-domain benchmark that supports evaluation under the CC-ICL setting. By employing a dual-level data hierarchy, CC-Bench enables legitimate ground-truth variation conditioned on the active criterion even when the task remains fixed. Experiments on CC-Bench reveal that most models exhibit a rigid boundary bias, struggling to align their decisions with the latent criterion. We also find that even a simple multi-criterion training strategy can significantly reduce this bias, improving Criterion Sensitivity and enabling 7B-scale models to surpass proprietary models without degrading general multimodal performance.

## 1. Introduction

In-context learning (ICL) has become a central paradigm for large language models (LLMs) and vision-language models (VLMs), enabling models to perform new tasks at inference time by conditioning on a few-shot support set without parameter updates (Alayrac et al., 2022; Awadalla et al., 2023; Laurençon et al., 2023; Yang et al., 2025; Bai et al., 2025b; Wang et al., 2025). This inference-time characterization is agnostic to how ICL capability is acquired, as prior work has shown it can be elicited through post-training (Chen et al., 2025b; Jia et al., 2025; Dou et al., 2026). Yet existing work (Jia et al., 2025; Li et al., 2026; Baldassini et al., 2024; Chen et al., 2025a) predominantly views ICL as a mechanism for **task induction**: the support set serves to resolve uncertainty about what the task is, assuming that once the task is induced (e.g., an addition task in Figure 1), its decision criterion remains fixed.

However, beyond learning new tasks, ICL can also serve another underexplored yet important purpose: **criterion adaptation** under fixed task semantics. In real-world applications, many tasks exhibit a stable high-level intent, yet their decision criteria are highly situation-dependent, varying with cultural norms, industry standards, or individual preferences. As shown in Figure 1, in Industrial Inspection, the tolerance of surface scratches can vary under different criteria. In such scenarios, few-shot support sets do not define new tasks; instead, they serve to calibrate the model's decision boundary, clarifying how a pre-existing concept should be judged under the current criterion.

Despite the practical importance of criterion adaptation, this capability is largely overlooked by current evaluations of multimodal ICL. We attribute this absence to two main factors: (i) **limited evaluation setting**, where the common practice of evaluation follows a "single-trial" ICL setting. By testing a model only once per query with a fixed (often randomly sampled) support set, this approach yields a static snapshot of performance rather than an assessment of dynamic adaptation; and (ii) **criterion-static benchmark design**, where existing benchmarks are not structured to mimic criterion shifts. General multimodal benchmarks (e.g., MMBench (Liu et al., 2024b), MMStar (Chen et al., 2024)) consist of image-question pairs that already form self-

---
[*]Equal contribution  [1]University of Science and Technology of China, Hefei, China [2]Megvii Technology Inc., Beijing, China. Correspondence to: Wei Ge <gewei@megvii.com>, Jikai Wang <wangjk@ustc.edu.cn>.

*Proceedings of the $43^{rd}$ International Conference on Machine Learning*, Seoul, South Korea. PMLR 306, 2026. Copyright 2026 by the author(s).

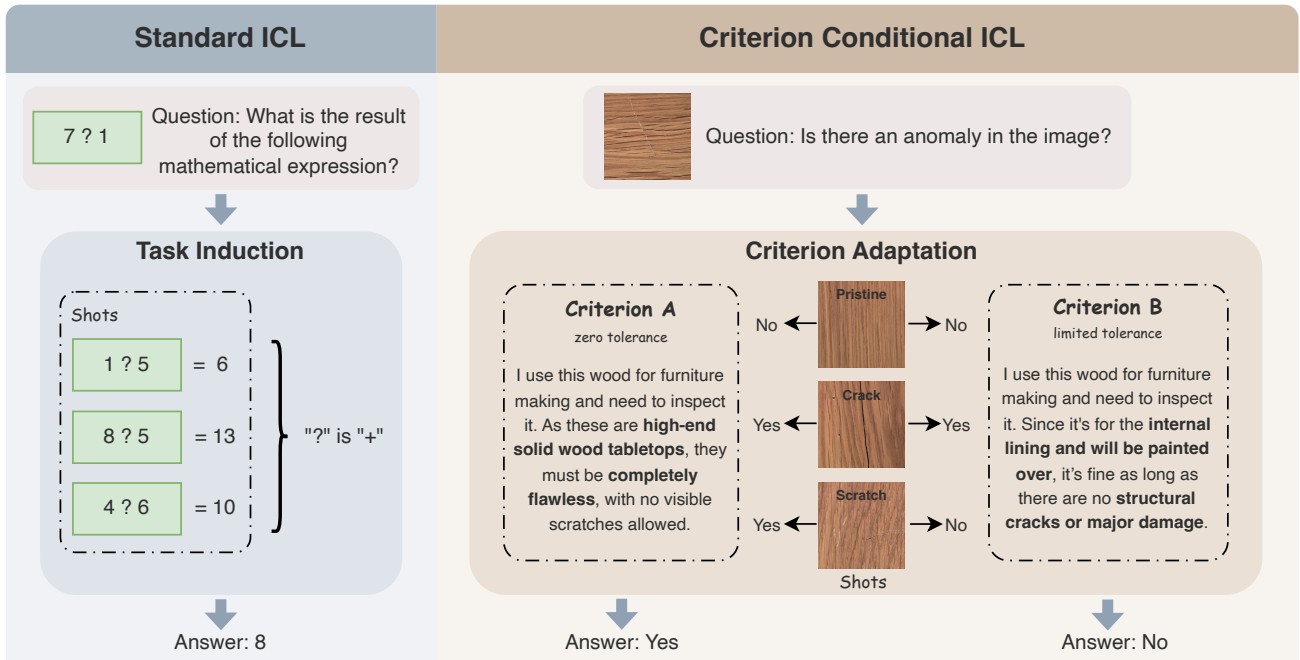

*Figure 1.* Standard ICL vs. Criterion-Conditional ICL (CC-ICL). In Standard ICL (left), the model primarily performs task induction to infer a single, invariant decision rule (e.g., identifying the "?" operator) from the support set. In contrast, CC-ICL (right) requires criterion adaptation: while the high-level task remains constant (e.g., anomaly detection), the model must recalibrate its decision boundaries based on context-specific criteria (Criterion A vs. B), leading to divergent predictions for the same query.

contained, unambiguous tasks with criterion-independent ground-truth answers. ICL-specific benchmarks (e.g., True-MICL (Chen et al., 2025b), VL-ICL (Zong et al., 2025)) deliberately under-specify the query and require the model to infer the task itself from the support set, which reflects shifts in task semantics rather than decision criteria. Neither category is designed to support settings where the decision criteria can legitimately shift while task semantics and high-level intent remain fixed.

To better evaluate criterion adaptation capabilities of VLMs, we first formalize a new setting, namely **Criterion-Conditional In-Context Learning (CC-ICL)**. Unlike standard ICL, CC-ICL curates multiple support sets for every query to mimic different criteria under fixed task semantics. A proficient model must exhibit two core competencies: adaptability in recalibrating boundaries for near-threshold samples, and robustness in maintaining consistent predictions for unaffected cases. To quantify these, we introduce two specialized metrics: **Criterion Sensitivity** and **Criterion Invariance**.

Furthermore, we introduce **CC-Bench**, a multi-domain benchmark utilizing a dual-level data hierarchy to enable controllable criterion shifts. Our systematic study of state-of-the-art VLMs reveals a prevalent rigid boundary bias, where models struggle to align decisions with context-conveyed criteria. To address this, we demonstrate that **Multi-Criterion Training (MCT)** can significantly en-

hance criterion adaptability in 7B-scale models without compromising robustness.

Our contributions are threefold:

- We propose CC-ICL for multimodal in-context learning that focuses on criterion adaptation, the ability to recalibrate decision boundaries under fixed task semantics. We also introduce two metrics to disentangle adaptability and robustness.

- We construct CC-Bench, a multi-domain benchmark with dual-level data hierarchy that enables controlled criterion shifts. This allows a systematic evaluation of decision criteria recalibration across image and video tasks.

- Through extensive experiments, we reveal a prevalent rigid boundary bias in state-of-the-art VLMs, and show that a simple multi-criterion training strategy substantially improves criterion adaptability in 7B-scale models with minimal loss of robustness.

## 2. Related Work

**In-Context Learning.** Originally popularized by GPT-3 (Brown et al., 2020), ICL has emerged as a cornerstone paradigm for large-scale models to perform new tasks without parameter updates (Zhang et al., 2022; Chowdhery et al.,

2023; Achiam et al., 2023; Grattafiori et al., 2024; Liu et al., 2024a; Yang et al., 2025). Recent advancements have successfully extended this capability into the multimodal domain, where VLMs demonstrate a remarkable aptitude for learning from interleaved vision-language exemplars (Alayrac et al., 2022; Awadalla et al., 2023; Laurençon et al., 2023; Zhao et al., 2024). Beyond empirical success, prior studies have investigated the underlying mechanism of ICL. Xie et al. (2022) posits that ICL reflects implicit Bayesian inference, while Akyürek et al. (2023) characterizes it as an algorithmic approximation of classical learning processes. Others have scrutinized the essential components of ICL, questioning the necessity of ground-truth labels (Min et al., 2022) or the genuine multimodality of the process (Chen et al., 2025a; Baldassini et al., 2024). Despite their differences, these approaches share a common assumption: the criterion used to assign labels remain the same across support sets. Under this paradigm, ICL is viewed primarily as a tool for recovering static task mappings, largely neglecting the potential for criterion-level adaptation.

**Evaluation of In-Context Learning.** With the growth of ICL capabilities, researchers have focused on building better benchmarks to measure this ability. Most studies (Jia et al., 2025; Li et al., 2026; Yang et al., 2024; Li et al., 2025) evaluate on general-purpose benchmarks, such as VQAv2 (Antol et al., 2015), TextVQA (Singh et al., 2019), OKVQA (Marino et al., 2019), MMStar (Chen et al., 2024), and MMBench (Liu et al., 2024b). In these datasets, few-shot examples merely serve as a mechanism for format alignment, and models may even overlook the visual modality (Baldassini et al., 2024; Chen et al., 2025a). To address this, recent works have introduced ICL-specific benchmarks such as VL-ICL (Zong et al., 2025) and True-MICL (Chen et al., 2025b). These benchmarks leave the query open-ended, forcing models to figure out the task themselves from the few-shot support set. Crucially, neither category accounts for settings in which the decision criterion shifts, even when task semantics remain the same. To bridge this gap, we introduce a novel benchmark for criterion adaptation, evaluating how effectively a model adjusts its predictions according to shifted decision criteria.

## 3. Criterion-Conditional In-Context Learning

### 3.1. Problem Formulation

**Standard In-Context Learning.** From a generative perspective, ICL can be interpreted as implicit inference over a latent concept $c$ shared across the context (Xie et al., 2022). Given a support set $\mathcal{S} = \{(x_i, y_i)\}_{i=1}^{N}$ and a query $x$, an autoregressive model produces predictions by conditioning on the entire context without updating its parameters $\theta$:

$$p_\theta(y \mid x, \mathcal{S}) = \int p_\theta(y \mid x, c)\, p_\theta(c \mid \mathcal{S})\, dc, \qquad (1)$$

While the latent concept $c$ in Equation (1) is defined broadly, in the setting of standard ICL, it is typically instantiated as a specific task $T$. Formally, a task $T$ can be characterized by the triplet $T = \{\mathcal{X}, \mathcal{Y}, \mathcal{M}\}$, where $\mathcal{X}$ denotes an input space, $\mathcal{Y}$ is an output space, and $\mathcal{M}$ denotes the set of mappings from $\mathcal{X}$ to $\mathcal{Y}$, which are governed by different criteria.

Within the scope of standard ICL, we usually consider tasks with $|\mathcal{M}| = 1$, where the mapping from $\mathcal{X}$ to $\mathcal{Y}$ is governed by a unique, unambiguous criterion. In such cases, Equation (1) can be discretized as:

$$p_\theta(y \mid x, \mathcal{S}) = \sum_{T \in \mathcal{T}} p_\theta(y \mid x, T)\, p_\theta(T \mid \mathcal{S}), \qquad (2)$$

where $\mathcal{T}$ denotes the task space, and $p_\theta(T \mid \mathcal{S})$ serves as **task induction**, identifying which task is exemplified by the support set.

As a consequence, standard ICL cannot represent scenarios in which multiple valid predictions arise from shifts in decision criteria rather than changes in the underlying task.

**Criterion-Conditional In-Context Learning.** CC-ICL generalizes standard ICL by considering tasks with $|\mathcal{M}| > 1$. In these tasks, task induction itself is insufficient to uniquely determine the label for a given query. The model must perform **criterion adaptation**, i.e., infer the active criterion from the support set and condition its predictions accordingly. In such cases, Equation (1) can be discretized as:

$$p_\theta(y \mid x, \mathcal{S}) = \sum_{T \in \mathcal{T}} \sum_{M \in \mathcal{M}} p_\theta(y \mid x, M) \atop p_\theta(M \mid T, \mathcal{S})\, p_\theta(T \mid \mathcal{S}), \qquad (3)$$

where $p_\theta(M \mid T, \mathcal{S})$ represents inferring the active criterion $M$ from the support set $\mathcal{S}$ given the task $T$, and predictions are then conditioned on the inferred $M$ via $p_\theta(y \mid x, M)$.

Under this setting, each query is evaluated under multiple support sets $\mathbb{S} = \{\mathcal{S}_M\}$, $\mathcal{S}_M = \{(x_i, y_i^M)\}_{i=1}^{N}$. All support sets share a common set of input samples $\mathcal{X}_{\mathcal{S}} = \{x_i\}_{i=1}^{N}$, while $y_i^M$ is independently generated according to a *criterion-conditioned* mapping of $M \in \mathcal{M}$.

### 3.2. Criterion-Conditional Metrics

Under the CC-ICL setting, a competent model is expected to exhibit two complementary capabilities: (i) criterion sensitivity, i.e., adjusting its predictions according to the criterion induced by the support set; and (ii) criterion invariance, i.e.,

maintaining consistent predictions when the ground-truth label is independent of the criterion.

**Criterion Invariance.** Criterion Invariance of the model can be defined as:

$$\text{CI} = \frac{1}{|\mathcal{X}_{\text{inv}}|} \sum_{x \in \mathcal{X}_{\text{inv}}} \prod_{M \in \mathcal{M}} \mathbb{I}\big[\arg\max_y p_\theta(y|x, S_M) = y^M\big],$$

(4)

where $y^M$ is the ground-truth label of $x$ under the criterion-conditioned mapping $M$, and $\mathcal{X}_{\text{inv}} = \{x \mid y^M = y^{M'}, \forall M, M' \in \mathcal{M}\}$ denotes the set of criterion-invariant samples.

A higher CI indicates that the model does not alter its predictions in response to irrelevant criterion shifts.

**Criterion Sensitivity.** Criterion Sensitivity of the model is defined as:

$$\text{CS} = \frac{1}{|\mathcal{X}_{\text{sen}}|} \sum_{x \in \mathcal{X}_{\text{sen}}} \prod_{M \in \mathcal{M}} \mathbb{I}[\arg\max_y p_\theta(y|x, S_M) = y^M],$$

(5)

where $y^M$ is the ground-truth label of $x$ under mapping $M$ and $\mathcal{X}_{\text{sen}} = \{x \mid y^M \neq y^{M'}, \exists M, M' \in \mathcal{M}\}$ denotes the set of criterion-sensitive samples.

A higher CS indicates that the model successfully adapts its decision boundary to the latent criterion specified by the support set.

## 4. Criterion-Conditional Dataset

### 4.1. Dataset Design

We construct **CC-Bench**, a multi-domain benchmark specifically designed to evaluate VLMs in the CC-ICL setting. It comprises $3,580$ samples, alongside a separate training set of $1,806$ samples. To systematically control criterion shifts under this setting, we implement a dual-level data hierarchy that organizes each domain into a two-tier logical structure: (1) **Category** represents the target entity (e.g., industrial products, medical organs or surveillance scenes). It defines the scope of the task, providing the stable task semantics needed for the model to ground its predictions. (2) **Sub-category** represents fine-grained state attributes (e.g., specific defect types, pathological conditions, or behavioral patterns). These serve as the atomic elements used to construct criteria, a nuanced dimension of judgment that is largely absent from existing ICL benchmarks.

This hierarchical design is specifically tailored to simulate criterion shift within a unified task. By remapping specific sub-categories to different binary labels, we can systematically construct test sets that reflect varying criteria, forcing the model to adapt its decision boundary according to the support set.

### 4.2. Data Sources and Preprocessing

Following our hierarchical design, we curate data from several representative sources to populate the three domains.

**Industrial Inspection.** We select 13 objects and textures from the MVTec Anomaly Detection dataset (Bergmann et al., 2021) as categories, excluding Toothbrush and Transistor, as their simplified anomaly distributions provide insufficient complexity for criterion shifts. The original fine-grained anomaly annotations are retained as sub-categories, covering a spectrum from severe structural defects to subtle surface irregularities.

**Medical Diagnosis.** We utilize RadImageNet-VQA (Butsanets et al., 2026), treating anatomical regions as categories and pathology types with varying severity as sub-categories. To ensure visual consistency between query and support sets, we manually annotate imaging views (e.g., axial, coronal, or sagittal) for each region. We filter out Shoulder and Hip where distinct, finite views could not be reliably identified, ensuring that criterion adaptation is not confounded by drastic viewpoint changes.

**Video Surveillance.** We combine UCF-Crime (Sultani et al., 2018) and NWPU Campus (Cao et al., 2023), treating all videos as a single category and specific behavioral patterns as sub-categories. Since UCF-Crime lacks frame-level labels and NWPU lacks anomaly annotations, we employ the Seed1.8 (Seed, 2025a) model for initial temporal grounding followed by rigorous manual refinement and full annotation of sampled anomaly clips. All filtered clips are uniformly sampled into 16-frame image sequences.

Detailed lists of categories and sub-categories for all domains are provided in Appendix A.1.

### 4.3. Criterion-Conditional Labeling

To systematically manifest criterion shifts, we establish a formal gradation of severity for sub-categories within each category. Specifically, we first leverage advanced LLMs (e.g., GPT-4o (Hurst et al., 2024)) to perform an initial ranking of sub-categories based on their perceived severity. These rankings are subsequently validated and refined by domain experts, yielding a spectrum of attributes, ranging from normal states and minor irregularities (e.g., surface scuffs or climbing fence) to high-priority risks (e.g., structural cracks or explosion).

Based on this severity spectrum, we define two distinct decision criteria across all categories, as illustrated in Figure 2. **Criterion A** adopts a "zero-tolerance" policy, where any deviation from the normal, pristine state is strictly classified as *Abnormal*. Under this criterion, even minor surface irregularities or early-stage pathological signals are mapped to abnormal. In contrast, **Criterion B** emphasizes functional

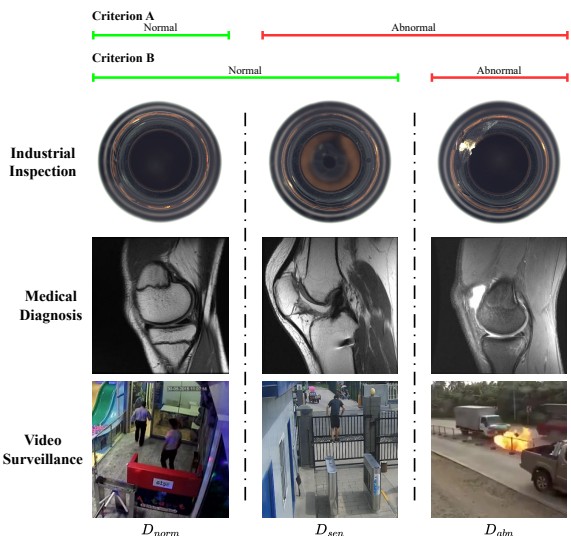

*Figure 2.* Illustration of Criterion-Conditional Labeling. The most typical frame within the video clip is used to represent the full clip in Video Surveillance.

integrity and high-priority risks. Here, the decision criterion shifts to categorize only significant threats—such as structural failures, life-threatening malignancies, or violent criminal incidents—as *Abnormal*, while minor deviations are remapped to the *Normal* label.

This remapping mechanism naturally categorizes our data into two functional groups, aligning with the formal definitions provided in Section 3.2. Specifically, we have (1) criterion-invariant samples: sub-categories that retain the same label under both criteria, such as pristine samples (always *Normal*) or catastrophic failures (always *Abnormal*). These samples anchor the task's semantic core. (2) criterion-sensitive samples: sub-categories whose labels flip between Criterion A and B (e.g., a minor scratch). These samples are the crux of our benchmark, as they explicitly test the model's ability to adapt its judgment based on the provided support set rather than relying on static task priors.

### 4.4. Dual-Context Pairing Strategy

To enable models to perceive shifted decision criteria, we propose a context pairing strategy that constructs paired support sets for each query. Following definitions in Section 4.3, the dataset is partitioned into three mutually exclusive subsets: (i) the always-abnormal set $\mathcal{D}_{abn}$ and (ii) the always-normal set $\mathcal{D}_{norm}$, which together constitute criterion-invariant samples, and (iii) the criterion-sensitive set $\mathcal{D}_{sen}$ comprising samples whose labels flip depending on the active criterion.

For each evaluation instance, we construct paired support

sets $\{\mathcal{S}_A, \mathcal{S}_B\}$ to explicitly manifest the shift between Criterion A and Criterion B. Each set is formed by a shared triplet of instances $\{s_{abn}, s_{norm}, s_{sen}\}$ randomly sampled from their respective subsets. Crucially, since labels of $s_{abn}$ and $s_{norm}$ remain fixed across criteria, $\mathcal{S}_A$ and $\mathcal{S}_B$ differ only in the label of $s_{sen}$, as shown in Figure 2.

This design forces the model to move beyond static task-level priors and instead dynamically calibrate its decision boundary based on the specific contextual cues provided by the label of $s_{sen}$.

### 4.5. Benchmark Splits and Statistics

To evaluate the model's ability to generalize decision criteria to unseen scenarios, we adopt an inter-category train-test split.

For the Industrial Inspection and Medical Diagnosis domains, we randomly sample two categories for training. For Video Surveillance, which contains a single category, we split the data at the sub-category level by randomly selecting six sub-categories for training.

Beyond categorical partitioning, we meticulously adjust the sample composition within the test benchmark. Specifically, a nearly $1 : 1$ ratio is maintained between criterion-sensitive samples ($\mathcal{D}_{sen}$) and criterion-invariant samples ($\mathcal{D}_{abn} \cup \mathcal{D}_{norm}$) across all categories. This balanced structure prevents the evaluation metrics from being dominated by a single sample type, thereby providing a nuanced reflection of the model's criterion adaptation stability.

Refer to Appendix A.1 for detailed category assignments.

## 5. Experiments

### 5.1. Experimental Setup

**Benchmarks.** We evaluate performance mainly on two kinds of benchmarks: (1) *CC-Bench*, our proposed suite spanning Industrial Inspection, Medical Diagnosis, and Video Surveillance. Beyond standard overall accuracy (Ov.), we report CI and CS as defined in Section 3.2, which specifically quantify a model's robustness and adaptability under shifting criteria. (2) *General Multimodal Benchmarks*, which evaluate general performance on MMBench (Liu et al., 2024b) and MMStar (Chen et al., 2024) to ensure that multi-criterion training does not degrade fundamental capabilities.

**Models.** We evaluate a diverse range of state-of-the-art VLMs: *Proprietary Models* include GPT-4o (Hurst et al., 2024) and Gemini 2.5 Pro (Comanici et al., 2025); *Open-Source Models* include Qwen2.5-VL-Instruct (7B/32B) (Bai et al., 2025b), Qwen3-VL-8B-Instruct (Bai et al., 2025a), InternVL-3.5-8B (Wang et al., 2025) and LLaVA-

*Table 1.* Performance of base VLMs on the CC-Bench. CS and CI denote Criterion Sensitivity (%) and Criterion Invariance (%), respectively. Ov. represents the average accuracy under each domain. Results reveal a consistent imbalance between invariance and sensitivity across models, indicating a rigid decision boundary bias under criterion shifts.

| Model | Industrial Inspection | | | Medical Diagnosis | | | Video Surveillance | | |
|---|---|---|---|---|---|---|---|---|---|
| | CS | CI | Ov. | CS | CI | Ov. | CS | CI | Ov. |
| *Proprietary API Models* | | | | | | | | | |
| GPT-4o | 29.74 | 82.46 | 58.00 | 9.76 | 34.19 | 22.13 | 8.57 | 70.95 | 39.76 |
| Gemini 2.5 Pro | 20.04 | 88.99 | 57.00 | 3.38 | 60.69 | 32.41 | 8.10 | 87.14 | 47.62 |
| *Open-Source Models* | | | | | | | | | |
| Qwen2.5-VL-7B-Instruct | 13.79 | 39.74 | 27.70 | 8.54 | 31.35 | 20.09 | 0.00 | 28.57 | 14.29 |
| Qwen2.5-VL-32B-Instruct | 17.67 | 58.58 | 39.60 | 14.26 | 35.10 | 24.81 | 2.86 | 39.52 | 21.19 |
| Qwen3-VL-8B-Instruct | 11.21 | 83.02 | 49.70 | 13.23 | 53.47 | 33.61 | 3.81 | 90.95 | 47.38 |
| InternVL3.5-8B | 16.81 | 70.15 | 45.40 | 9.85 | 41.04 | 25.65 | 14.76 | 38.10 | 26.43 |
| LLaVA-OneVision-1.5-8B-Instruct | 15.95 | 38.81 | 28.20 | 7.50 | 29.43 | 18.61 | 17.14 | 50.95 | 34.05 |

OneVision-1.5-8B-Instruct (An et al., 2025). For all experiments, we set the sampling temperature to 0 to ensure deterministic and reproducible outputs.

**Multi-Criterion Training.** To overcome the limitations of static decision-making, we introduce MCT, a strategy designed to foster criterion adaptation capabilities. MCT leverages the training split described in Section 4.5, with models trained and evaluated separately on each domain. , following the pairing protocol in Section 4.4, each shared triplet yields two support sets that contain identical images but differ only in the label assigned to the criterion-sensitive instance. For each triplet, we then construct two training instances by combining the same query with each of the two support sets. As a result, the model is trained to produce different answers to the same query when conditioned on different support sets, ensuring that supervision is criterion-conditioned rather than label-static. This extends the model's task induction behavior by enabling flexible criterion adaptation.

**Implementation Details.** Our MCT implementation is built upon the `swift` framework. For efficient fine-tuning, we employ LoRA (Hu et al., 2022) with a rank $r = 4$ and $\alpha = 32$. Models are trained for 2 epochs with a learning rate of $1 \times 10^{-4}$ and a batch size of 4, on NVIDIA A100 (80GB) GPUs. Unless otherwise specified, evaluations use a 3-shot support set. For a comprehensive list of hyperparameters and data preprocessing details, please refer to Appendix B.1.

**5.2. Analysis of Base Models**

We first evaluate state-of-the-art VLMs under the CC-ICL setting to establish a baseline. As shown in Table 1, there is a severe imbalance between CI and CS across all evaluated models and domains. Specifically, while models achieve moderate to high CI, their CS is consistently marginal,

with most models scoring below 20%. Notably, the CS of Qwen2.5-VL-7B-Instruct in the Video Surveillance domain even drops to 0.00%, reflecting a complete failure in criterion adaptation.

This performance gap suggests that base models are dominated by static task-level priors. Instead of performing criterion adaptation from the provided support set, models tend to rely on a fixed task induction pattern inherited during pre-training. For instance, in Industrial Inspection, models maintain relatively high CI but fail to recalibrate decisions for criterion-sensitive samples. We characterize this asymmetry between performance on invariant and sensitive samples as a *rigid boundary bias*.

Furthermore, results underscore that criterion adaptation is not an emergent property of base models. Existing VLMs across both proprietary and open-source families lack the fundamental mechanism required to dynamically recalibrate decision boundaries based on shifting contextual cues. This systematic gap necessitates an explicit training intervention, such as our proposed MCT, to extend models beyond rigid task induction to a flexible criterion adaptation paradigm.

**5.3. Mitigating Rigid Boundary Bias via MCT**

As shown in Table 2, applying MCT leads to substantial performance gains across all evaluated models and domains. These improvements are primarily driven by the substantial gains in CS, which consistently exceed the improvements in CI. For instance, Qwen2.5-VL-7B-Instruct achieves a remarkable $+74.57\%$ increase in CS within the Medical Diagnosis domain, compared to a more modest $+18.74\%$ rise in CI. Notably, a 7B-scale model fine-tuned with MCT can achieve overall accuracy comparable to or even exceeding much larger open-source models and strong proprietary models.

*Table 2.* Evaluation of MCT's effectiveness. $\Delta$ represents the performance difference after MCT tuning compared to the base model. Finally, avg($\Delta$) summarizes the average improvement across all evaluated models.

| Model | Industrial Inspection | | | Medical Diagnosis | | | Video Surveillance | | |
|---|---|---|---|---|---|---|---|---|---|
| | CS | CI | Ov. | CS | CI | Ov. | CS | CI | Ov. |
| *Qwen2.5-VL-7B-Instruct* | | | | | | | | | |
| Base | 13.79 | 39.74 | 27.70 | 8.54 | 31.35 | 20.09 | 0.00 | 28.57 | 14.29 |
| MCT | 68.32 | 74.63 | 71.70 | 83.11 | 50.09 | 66.39 | 45.71 | 81.90 | 63.81 |
| $\Delta$ | +54.53 | +34.89 | +44.00 | +74.57 | +18.74 | +46.30 | +45.71 | +53.33 | +49.52 |
| *Qwen3-VL-8B-Instruct* | | | | | | | | | |
| Base | 11.21 | 83.02 | 49.70 | 13.23 | 53.47 | 33.61 | 3.81 | 90.95 | 47.38 |
| MCT | 69.40 | 69.22 | 69.30 | 72.51 | 61.88 | 67.13 | 32.38 | 80.95 | 56.67 |
| $\Delta$ | +58.19 | -13.80 | +19.60 | +59.28 | +8.41 | +33.52 | +28.57 | -10.00 | +9.29 |
| *InternVL-3.5-8B* | | | | | | | | | |
| Base | 16.81 | 70.15 | 45.40 | 9.85 | 41.04 | 25.65 | 14.76 | 38.10 | 26.43 |
| MCT | 36.64 | 72.57 | 55.90 | 67.45 | 62.07 | 64.72 | 36.19 | 75.24 | 55.71 |
| $\Delta$ | +19.83 | +2.42 | +10.50 | +57.60 | +21.03 | +39.07 | +21.43 | +37.14 | +29.28 |
| *LLaVA-OneVision-1.5-8B-Instruct* | | | | | | | | | |
| Base | 15.95 | 38.81 | 28.20 | 7.50 | 29.43 | 18.61 | 17.14 | 50.95 | 34.05 |
| MCT | 56.25 | 71.93 | 64.60 | 80.58 | 48.99 | 64.58 | 29.05 | 87.62 | 58.33 |
| $\Delta$ | +40.30 | +33.12 | +36.40 | +73.08 | +19.56 | +45.97 | +11.91 | +36.67 | +24.28 |
| avg($\Delta$) | +43.21 | +14.16 | +27.63 | +66.13 | +16.94 | +41.22 | +26.91 | +29.29 | +28.10 |

Experimental results demonstrate that MCT effectively fosters adaptability without sacrificing fundamental robustness. In specific cases where CI exhibits localized reductions, such as Qwen3-VL-8B-Instruct in the industrial and video domains, the substantial leap in CS (+58.19% and +28.57%) still leads to a significant net increase in overall accuracy. Such a trade-off indicates that MCT does not indiscriminately degrade model stability but rather relaxes excessively rigid decision boundaries to prioritize flexible criterion adaptation.

Overall, these results show that MCT fundamentally reshapes the mechanism through which models interpret support sets. Extended beyond a static task induction pattern to a dynamic criterion adaptation mechanism, trained models successfully mitigate the rigid boundary bias prevalent in base models. These findings confirm that explicit multi-criterion supervision is a highly effective path toward eliciting criterion adaptation in VLMs, enabling them to recalibrate decision boundaries based on contextual cues.

### 5.4. Chain-of-Thought Analysis

To further examine the behavioral differences between base and MCT-trained models, we prompt them to generate reasoning traces alongside predictions. We observe a clear distinction: the base model reasons purely from the query image without any reference to the support set, whereas the MCT-trained model explicitly performs visual comparisons between the query and each support image, grounding its prediction in the visually demonstrated criterion rather than

abstract textual patterns (shown in Appendix C.1, Figure 6).

- **Base model:** "Shape: The nut appears to be roughly spherical... Surface Texture: The surface has a natural, slightly rough texture... Given these observations, there does not appear to be any anomaly in the image... Final answer: No."

- **MCT-trained model:** "Compare the image to the first image: The first image shows a hazelnut with a small crack on its surface. Compare the image to the second image: The second image shows a hazelnut without any visible cracks... The fourth image has a similar anomaly to the third example. Therefore, the fourth image also has an anomaly."

This directly demonstrates that MCT elicits criterion-conditioned visual reasoning rather than criterion-agnostic prediction mapping.

### 5.5. Ablation Study

We further conduct a set of ablation studies to examine the generalization of MCT and to investigate the factors underlying the limited criterion adaptation in base models.

**Ablation 1: Cross-Domain Generalization of Criterion Adaptation.** To evaluate the domain-agnostic nature of MCT, we train Qwen2.5-VL-7B-Instruct on a single domain and evaluate its adaptation performance on the remaining unseen domains in Table 3. Several key findings can be drawn from these cross-domain results.

*Table 3.* Cross-domain generalization of MCT models. Models tuned with MCT on a single domain are evaluated on the remaining domains to assess cross-domain generalization. Accuracy on general multimodal benchmarks, MMBench (MMB) and MMStar (MMS), are additionally reported to verify that domain-specific MCT tuning does not degrade general multimodal capabilities.

| Model | Industrial Inspection | | | Medical Diagnosis | | | Video Surveillance | | | General | |
|---|---|---|---|---|---|---|---|---|---|---|---|
| | CS | CI | Ov. | CS | CI | Ov. | CS | CI | Ov. | MMB | MMS |
| *Qwen2.5-VL-7B-Instruct* | | | | | | | | | | | |
| Base | 13.79 | 39.74 | 27.70 | 8.54 | 31.35 | 20.09 | 0.00 | 28.57 | 14.29 | 87.57 | 59.73 |
| MCT (Industrial) | 68.32 | 74.63 | 71.70 | 57.13 | 49.73 | 53.38 | 14.76 | 33.81 | 24.29 | 87.73 | 60.20 |
| MCT (Medical) | 64.66 | 57.09 | 60.60 | 83.11 | 50.09 | 66.39 | 3.33 | 29.05 | 16.19 | 87.66 | 60.67 |
| MCT (Surveillance) | 46.12 | 65.11 | 56.30 | 49.35 | 50.65 | 37.96 | 45.71 | 81.90 | 63.81 | 87.32 | 60.27 |

*MCT enables cross-domain criterion adaptation.* Models trained on a specific domain exhibit substantial improvements when tested on unseen areas. For instance, the model trained in the industrial domain improves CS in the medical domain from 8.54% to 57.13%. This indicates that MCT teaches the model the underlying meta-logic of how to adapt criteria rather than merely memorizing domain-specific patterns, thereby mitigating the rigid boundary bias in most unseen scenarios.

*Modality gaps constrain transferability.* A notable exception exists in the transfer from static to dynamic domains. While models separately trained in medical and industrial domains generalize well to each other, their performance in Video Surveillance remains nearly stagnant at 3.33% and 14.76% respectively. This stagnation suggests that models trained on static images struggle to recalibrate decision boundaries for tasks requiring temporal context. Conversely, the model trained in the video domain exhibits robust generalization to static images, with CS reaching 46.12% in Industrial Inspection and 49.35% in Medical Diagnosis. This suggests that MCT on video data enables a "dimension-reduction" generalization where the model applies sophisticated temporal induction skills to simpler static visual criteria.

*General multimodal capabilities remain preserved.* Table 3 shows that MCT does not degrade core performance. Scores on general benchmarks like MMBench and MMStar remain stable across all training settings. This confirms that MCT elicits criterion adaptation without causing catastrophic forgetting of pre-trained knowledge.

**Ablation 2: Multi-criterion Generalization.** To further investigate whether MCT learns genuine criterion adaptation rather than memorizing fixed criteria, we extend the standard two-criteria setting by introducing an additional "medium-tolerance" criterion, forming a more fine-grained spectrum of decision boundaries.

Models trained under the two-criteria setting are directly evaluated in this three-criteria setting without additional fine-tuning. As shown in Table 4, CS remains largely stable compared to the two-criteria setting, indicating strong

generalization to previously unseen decision boundaries. These results suggest that MCT induces a flexible criterion adaptation capability rather than memorizing fixed decision rules.

Additional experiments in Appendix C.2 further show that training and evaluating under the three-criteria setting substantially improves CS (e.g., +46.21% over the baseline in Industrial Inspection domain) while preserving strong transferability across domains and criterion configurations.

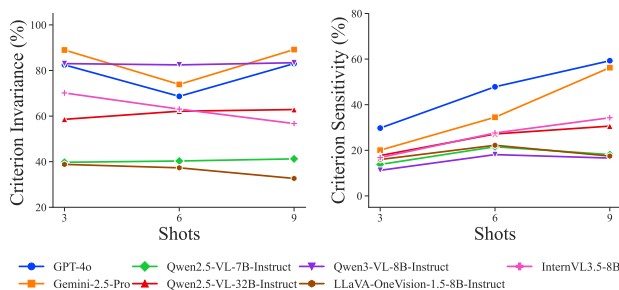

*Figure 3.* Effect of the number of shots on model performance (Industrial Inspection). The left panel reports CI (%) scores, while the right panel reports CS (%) scores. CS improvements remain marginal and inconsistent, with additional shots yielding gains primarily for proprietary models and having minimal impact on open-source ones.

**Ablation 3: Effect of the Number of Shots on Criterion Adaptation.** We investigate whether criterion adaptation in base models can be guided by increasing the number of shots. To this end, we evaluate base models under 3-shot, 6-shot, and 9-shot settings.

As shown in Figure 3, the effect on CS is inconsistent and generally limited. While some proprietary models show noticeable CS increases with more shots, such as GPT-4o on Industrial Inspection (29.74% to 59.27%), open-source models remain largely insensitive. For instance, Qwen2.5-VL-7B-Instruct exhibits only marginal CS changes on Industrial Inspection (13.79% to 18.10%). Results on the Medical Diagnosis and Video Surveillance domains exhibit similar trends and are reported in Appendix C.3.

*Table 4.* Generalization from 2-criteria training to 3-criteria evaluation. Models tuned with MCT under the 2-criteria setting on a single domain are evaluated with 3-criteria on both the same and the remaining domains.

| Model | Industrial Inspection | | | Medical Diagnosis | | | Video Surveillance | | |
|---|---|---|---|---|---|---|---|---|---|
| | CS | CI | Ov. | CS | CI | Ov. | CS | CI | Ov. |
| *Qwen2.5-VL-7B-Instruct* | | | | | | | | | |
| Base | 12.12 | 44.07 | 27.20 | 3.75 | 29.34 | 16.71 | 0.00 | 28.57 | 14.29 |
| MCT (Industrial) | 55.60 | 66.79 | 61.60 | 50.28 | 33.73 | 41.90 | 10.95 | 31.43 | 21.19 |
| MCT (Medical) | 45.47 | 48.32 | 47.00 | 63.98 | 32.18 | 47.87 | 0.95 | 28.57 | 14.76 |
| MCT (Surveillance) | 41.59 | 54.66 | 48.60 | 29.92 | 23.58 | 26.71 | 42.38 | 66.67 | 54.52 |

Overall, these results show that the rigid boundary bias is not simply a consequence of an insufficient number of shots. While proprietary models exhibit a degree of criterion adaptability that scales with the number of shots, open-source models remain trapped in static task-level priors. This disparity suggests that reliable criterion adaptation requires explicit criterion-aware training rather than relying on context scaling. Our proposed MCT provides this necessary supervision, enabling models to move beyond rigid task induction toward a flexible, conditioning-based decision mechanism.

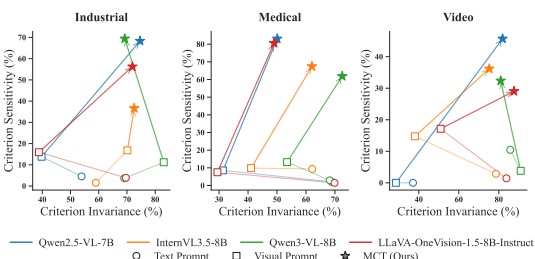

*Figure 4.* Effect of prompt modality and MCT on criterion adaptation across domains. For each model, visual prompts and text prompts are compared, together with their MCT-trained counterparts. Different colors denote different models.

**Ablation 4: Comparison of Visual and Text Prompts.**
We examine whether base models can acquire criterion adaptation through prompt modality alone. Across all three CC-Bench domains, replacing visual prompts with text prompts consistently shifts model behavior rightward in the CS–CI plane, characterized by increased CI but unchanged or even decreased CS, as shown in Figure 4. This pattern indicates that text-only prompting does not enable criterion adaptation, but instead reinforces rigid decision boundaries.

In contrast, models trained with MCT move to the top right across all domains, achieving substantially higher CS compared to both visual- and text-prompted base counterparts. These results demonstrate that criterion adaptation in the CC-ICL setting cannot be obtained via prompt engineering alone, and requires explicit criterion-aware training.

## 6. Conclusion

This paper formalizes CC-ICL, a novel setting that decouples decision criteria from task semantics to move beyond the "one-task, one-criterion" assumption in standard ICL. Through our proposed CC-Bench, we identify a prevalent rigid boundary bias in state-of-the-art VLMs, which rely on static task-level priors rather than performing dynamic criterion adaptation based on contextual cues. To bridge this gap, we introduce MCT, a strategy that effectively extends models beyond static task induction to flexible criterion adaptation. Experimental results show that our MCT-enhanced 7B model significantly mitigates this decision bias, rivaling strong proprietary systems in criterion-sensitive reasoning while preserving general multimodal ability. We believe that CC-ICL represents a fundamental step toward achieving more versatile and human-aligned AI, where the ability to perceive and adapt to underlying criteria is as critical as mastering the task itself.

## Acknowledgement

Wei Ge led this project. And this work was supported by Anhui Provincial Natural Science Foundation under Grant 2508085MF144.

## Impact Statement

This work presents a methodological study of CC-ICL, introducing a new evaluation setting, together with a benchmark and metrics, to analyze how VLMs adjust their decision boundaries under shifting criteria. All data used are from publicly available and open-source datasets, with no new data collected.

By revealing rigid boundary bias under criterion shifts, this work helps prevent overestimation of models' ability to adapt decision boundaries. The proposed benchmark serves as a diagnostic evaluation tool and does not introduce new societal risks beyond those inherent to existing VLMs.

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

# A. Dataset Details

## A.1. Details of Categories and Sub-categories

This section details the categories and sub-categories forming the CC-Dataset, as summarized in Table 5, 6, and 7.

As shown in Figure 5, for the training split we select the *cable* and *leather* categories from Industrial Inspection, the *abdomen* and *ankle-foot* categories from Medical Diagnosis, and a subset of sub-categories including *good*, *forgetting backpack*, *crossing lawn*, *littering*, *shooting*, and *arson* from Video Surveillance. For these training categories, we maintain a balanced distribution by ensuring the number of samples per sub-category remains roughly equal. This design prevents the model from developing frequency biases toward specific defect types or pathological states, forcing it to focus on the logical mapping between the provided demonstrations and the query instance.

The benchmark is constructed from the remaining categories to evaluate n-shot generalization to unseen tasks and environments. To rigorously assess the model's sensitivity to criterion shifts, we employ a controlled sampling strategy maintaining an approximate $1:1$ ratio between CI and CS samples, as shown in Figure 5. This distribution is designed to neutralize any potential performance inflation caused by the model's inherent alignment with pre-trained label priors. By ensuring an equal proportion of invariant and sensitive samples, the aggregate metric serves as a reliable indicator of whether the model is genuinely adapting its decision boundary based on the provided support set, rather than simply relying on static, task-agnostic common sense.

*Table 5.* Categories and Sub-categories in the Medical Diagnosis Domain.

| | Category | Sub-category | Size | | Category | Sub-category | Size |
|---|---|---|---|---|---|---|---|
| | | osseous_neoplasm | 100 | | | osseous_neoplasm | 99 |
| | | renal_lesion | 100 | | | fat_containing_tumor | 100 |
| | | liver_lesion | 100 | | | osseous_disruption | 99 |
| | Abdomen | gallstone | 100 | | Ankle-Foot | Plantar_plate_tea | 99 |
| | | urolithiasis | 100 | | | atfl_pathology | 100 |
| | | post_op | 100 | | | cfl_pathology | 99 |
| CT | | normal | 100 | | | post_op | 99 |
| | | interstitial_lung_disease | 98 | | | normal | 100 |
| | | Parenchyma_destruction | 98 | MRI | | fracture | 100 |
| | Lung | Nodule | 96 | | | bone_inflammation | 100 |
| | | Bronchiectasis | 100 | | Knee | post_operative_acl | 77 |
| | | Airspace_opacity | 98 | | | fcl_pathology | 100 |
| | | normal | 100 | | | muscle_strain | 100 |
| | | cord_pathology_ | 75 | | | normal | 100 |
| | | dural_epidural_abn | 75 | | | acute_infarct | 49 |
| | | foraminal_pathology | 74 | | | arteriovenous_anomaly | 50 |
| MRI | Spine | cystic_lesions | 100 | | Brain | extra | 50 |
| | | disc_pathology | 100 | | | white_matter_change | 100 |
| | | facet_arthropathy | 96 | | | focal_flair_hyper | 99 |
| | | normal | 75 | | | normal | 50 |

*Table 6.* Categories and Sub-categories in the Video-level Domain.

| Category | Sub-category | Size | | Category | Sub-category | Size |
|---|---|---|---|---|---|---|
| | good | 80 | | | Scuffle | 50 |
| | Forgetting backpack | 15 | | | Stealing | 50 |
| | Crossing lawn | 14 | | | Vandalism | 30 |
| Video-level | Littering | 13 | | Video-level | Arrest | 30 |
| | Shooting | 15 | | | Robbery | 55 |
| | Arson | 15 | | | Road Accidents | 55 |
| | Jaywalking | 50 | | | Explosion | 40 |

*Table 7.* Categories and Sub-categories in the Industrial Inspection Domain.

| Category | | Sub-category | Size |
|---|---|---|---|
| Texture | Carpet | hole | 12 |
| | | cut | 12 |
| | | metal_contamination | 12 |
| | | color | 19 |
| | | thread | 19 |
| | | good | 12 |
| | Grid | broken | 7 |
| | | bent | 7 |
| | | metal_contamination | 8 |
| | | glue | 11 |
| | | thread | 11 |
| | | good | 8 |
| | Leather | cut | 19 |
| | | fold | 17 |
| | | poke | 18 |
| | | color | 19 |
| | | glue | 19 |
| | | good | 20 |
| | Tile | crack | 17 |
| | | glue_strip | 12 |
| | | rough | 12 |
| | | oil | 13 |
| | | gray_stroke | 13 |
| | | good | 33 |
| | Wood | hole | 10 |
| | | combined | 11 |
| | | liquid | 10 |
| | | scratch | 21 |
| | | color | 8 |
| | | good | 19 |
| Objects | Bottle | broken_large | 10 |
| | | broken_small | 10 |
| | | contamination | 21 |
| | | good | 10 |
| | Capsule | crack | 23 |
| | | poke | 21 |
| | | squeeze | 20 |
| | | scratch | 23 |
| | | faulty_imprint | 22 |
| | | good | 23 |

| Category | | Sub-category | Size |
|---|---|---|---|
| Objects | Cable | missing_cable | 12 |
| | | missing_wire | 10 |
| | | cut_inner_insulation | 14 |
| | | cut_outer_insulation | 10 |
| | | combined | 11 |
| | | cable_swap | 12 |
| | | bent_wire | 13 |
| | | poke_insulation | 10 |
| | | good | 15 |
| | Hazelnut | hole | 6 |
| | | crack | 6 |
| | | cut | 6 |
| | | print | 17 |
| | | good | 7 |
| | Metal nut | bent | 18 |
| | | flip | 18 |
| | | scratch | 23 |
| | | color | 22 |
| | | good | 18 |
| | Pill | crack | 17 |
| | | contamination | 17 |
| | | pill_type | 9 |
| | | combined | 17 |
| | | faulty_imprint | 19 |
| | | color | 25 |
| | | scratch | 24 |
| | | good | 17 |
| | Screw | manipulated_front | 14 |
| | | thread_top | 14 |
| | | thread_side | 14 |
| | | scratch_head | 24 |
| | | scratch_neck | 25 |
| | | good | 15 |
| | Zipper | broken_teeth | 11 |
| | | split_teeth | 11 |
| | | squeezed_teeth | 12 |
| | | combined | 12 |
| | | fabric_interior | 16 |
| | | fabric_border | 17 |
| | | rough | 17 |
| | | good | 12 |

# B. Experimental Setup Details

## B.1. Training Hyperparameter Configuration

To ensure consistency across all models' experiments, we strictly adhere to a unified hyperparameter configuration as listed in Table 8. We apply LoRA adapters to the self-attention modules across all Transformer layers of the language model. Specifically, the adaptation is targeted at the Query ($W_q$) and Key ($W_k$) projection matrices.

## B.2. Prompts

For all evaluations on general benchmarks and CC-Bench, prompts are constructed as a sequence of single-round user input, each comprising an image and a question–answer pair. Questions used for each domain of CC-Bench are shown in Table 9. An additional suffix is appended to each question for CC-Bench, in order to explicitly constrain the response format and

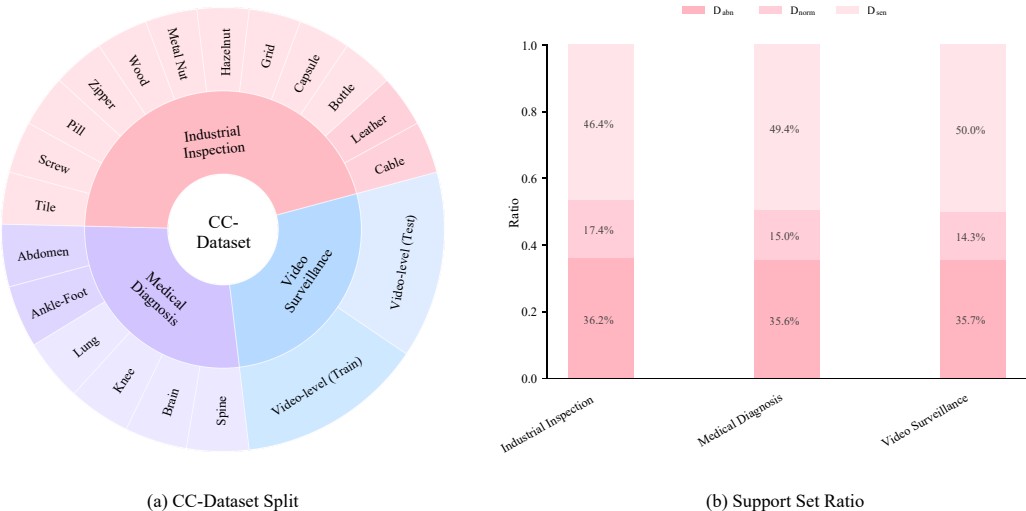

(a) CC-Dataset Split

(b) Support Set Ratio

*Figure 5.* Data Split and Statistics of CC-Bench. (a) CC-Dataset Split. Dark-colored sectors represent the training sets and light-colored sectors denote the test sets. (b) Support Set Ratio. The proportions of abnormal ($D_{abn}$), normal ($D_{norm}$), and sensitive ($D_{sen}$) samples within each domain.

*Table 8.* LoRA Fine-tuning Configuration.

| Parameter | Value |
|---|---|
| Batch size per device | 2 |
| Gradient accumulation steps | 2 |
| Epochs | 2 |
| Learning rate | 1e-4 |
| Warmup ratio | 0.05 |
| Optimizer | AdamW |
| Weight decay | 0.1 |
| Learning rate scheduler | Cosine |
| Precision | bfloat16 |
| Max sequence length | 2048 |
| Random seed | 42 |
| LoRA rank | 4 |
| LoRA alpha | 32 |

reduce potential confounding effects from prompt structure alignment. Full demonstrations for both prompts are listed below.

---

**MMBench/MMStar Visual Prompt Format**

```
System: You are a helpful assistant.
User:
  <image 1> Question: <question 1> Answer: <answer 1>
  ...
  <image N> Question: <question N> Answer: <answer N>
  <query image> Question: <query question> Answer:
```

---

**CC-Bench Visual Prompt Format**

```
System: You are a helpful assistant.
User:
  <image 1> Question: <question 1> Answer with yes/no. Answer: <answer 1>
  ...
  <image N> Question: <question N> Answer with yes/no. Answer: <answer N>
  <query image> Question: <query question> Answer with yes/no. Answer:
```

*Table 9.* Question Setup for CC-Bench.

| Domain | Question |
|---|---|
| Industrial Inspection | Is there an anomaly in the image? |
| Medical Diagnosis | Is this image abnormal? |
| Video Surveillance | Is there any abnormal behavior in this video? |

In Section 5.5, we investigated the impact of different prompt modalities on the criterion-adaptation performance of base models. To evaluate the model's response to pure text prompt, we adapted the CC-Bench Text Prompt Format by replacing visual examples with their corresponding sub-category labels. Instead of providing raw images as support sets, we supplied the model with their sub-category ground-truths.

**CC-Bench Text Prompt Format**

```
System: You are a helpful assistant.
User:
  If <sub-category 1> is in the image, question: <question 1> Answer with yes/no. An-
swer: <answer 1>
  ...
  If <sub-category N> is in the image, question: <question N> Answer with yes/no. An-
swer: <answer N>
  <query image> Question: <query question> Answer with yes/no. Answer:
```

## C. More Experimental Analysis

### C.1. Qualitative Analysis of Criterion-Conditioned Reasoning

As shown in Figure 6, we observe a consistent behavioral shift across both domains after MCT. The base model primarily relies on local visual attributes of the query image and tends to generate predictions without effectively incorporating contextual demonstrations. In contrast, the MCT-trained model explicitly performs support-grounded comparisons and derives predictions according to the demonstrated criterion. This observation aligns with the quantitative improvements in CS and provides additional evidence that MCT induces context-conditioned reasoning rather than memorizing static prediction rules.

### C.2. Extended Analysis under the Three-Criteria Setting

#### C.2.1. EFFECTIVENESS OF MCT UNDER THE THREE-CRITERIA SETTING

As shown in Table 10, MCT remains highly effective under the more challenging three-criteria setting, leading to substantial performance gains across all evaluated domains. Similar to the observations in the two-criteria setting, these improvements are primarily driven by significant increases in CS, demonstrating enhanced adaptability to shifted decision boundaries.

Specifically, Qwen2.5-VL-7B-Instruct achieves remarkable CS improvements of +46.21%, +53.00%, and +37.14% in Industrial Inspection, Medical Diagnosis, and Video Surveillance, respectively. In contrast, the corresponding CI improvements are relatively more moderate, indicating that MCT primarily strengthens criterion adaptation rather than merely reinforcing invariant predictions. Notably, despite the increased complexity introduced by the additional "medium-tolerance" criterion, the MCT-trained model maintains strong overall performance across all domains, achieving substantial gains in overall accuracy (+35.70%, +35.37%, and +41.66%). These results suggest that MCT can generalize beyond two-criteria

*Figure 6.* Comparison of reasoning traces before and after MCT.

configurations and effectively adapt to more fine-grained and continuous decision boundaries.

Overall, the results further support that MCT fundamentally reshapes how models interpret support sets under the CC-ICL setting. Rather than relying on static task-level priors, the trained model learns to dynamically infer criterion-dependent decision boundaries from contextual examples, thereby mitigating the rigid boundary bias observed in base models.

*Table 10.* Evaluation of MCT under 3-criteria setting. $\Delta$ denotes the improvement over the base model.

| Model | Industrial Inspection | | | Medical Diagnosis | | | Video Surveillance | | |
|---|---|---|---|---|---|---|---|---|---|
| | CS | CI | Ov. | CS | CI | Ov. | CS | CI | Ov. |
| *Qwen2.5-VL-7B-Instruct* | | | | | | | | | |
| Base | 12.12 | 44.07 | 27.20 | 3.75 | 29.34 | 16.71 | 0.00 | 28.57 | 14.29 |
| MCT | 58.33 | 68.01 | 62.90 | 56.75 | 47.53 | 52.08 | 37.14 | 74.76 | 55.95 |
| $\Delta$ | +46.21 | +23.94 | +35.70 | +53.00 | +18.19 | +35.37 | +37.14 | +46.19 | +41.66 |

## C.2.2. CROSS-DOMAIN GENERALIZATION UNDER THE THREE-CRITERIA SETTING

To further evaluate the generalization ability of MCT under the more challenging three-criteria setting, we train Qwen2.5-VL-7B-Instruct on a single domain and evaluate its adaptation performance on the remaining unseen domains, as shown in Table 11. Several important observations can be drawn from these cross-domain results.

*MCT retains cross-domain criterion adaptation under richer criterion structures.* Even under the more complex three-criteria setting with an additional "medium-tolerance" criterion, models trained on a specific domain still exhibit clear improvements on unseen domains. For example, the model trained on the Medical Diagnosis domain improves CS in Industrial Inspection from 12.12% to 52.46%, while the model trained on the Video Surveillance domain improves CS in Medical Diagnosis from

3.75% to 35.18%. These results suggest that MCT learns transferable criterion adaptation behaviors rather than memorizing domain-specific criterion mappings, and that such adaptation ability remains robust under richer criterion structures.

*Modality gaps continue to constrain transferability.* Consistent with the observations in the two-criteria setting, transferring from static-image domains to the Video Surveillance domain remains challenging. Models trained on Industrial Inspection and Medical Diagnosis achieve only 9.05% and 2.38% CS, respectively, when evaluated on Video Surveillance. This indicates that criterion adaptation learned from static images does not naturally transfer to tasks requiring temporal reasoning and dynamic context understanding.

In contrast, the model trained on the Video Surveillance domain exhibits substantially stronger reverse generalization to static-image domains, achieving 37.50% CS on Industrial Inspection and 35.18% on Medical Diagnosis. This asymmetric transfer pattern further supports the hypothesis that temporal criterion adaptation learned from video data can generalize to simpler static scenarios, whereas the reverse direction remains constrained by modality gaps.

Overall, these results demonstrate that MCT maintains strong cross-domain transferability even under the more complex three-criteria setting, further supporting that the learned capability reflects a generalized criterion adaptation mechanism rather than memorization of fixed decision rules.

*Table 11.* Cross-domain generalization of MCT under the 3-criteria setting. Models tuned with MCT on a single domain are evaluated on the remaining domains.

| Model | Industrial Inspection | | | Medical Diagnosis | | | Video Surveillance | | |
|---|---|---|---|---|---|---|---|---|---|
| | CS | CI | Ov. | CS | CI | Ov. | CS | CI | Ov. |
| *Qwen2.5-VL-7B-Instruct* | | | | | | | | | |
| Base | 12.12 | 44.07 | 27.20 | 3.75 | 29.34 | 16.71 | 0.00 | 28.57 | 14.29 |
| MCT (Industrial) | 58.33 | 68.01 | 62.90 | 16.91 | 45.85 | 38.19 | 9.05 | 31.90 | 20.48 |
| MCT (Medical) | 52.46 | 55.17 | 51.70 | 56.75 | 47.53 | 52.08 | 2.38 | 29.05 | 15.71 |
| MCT (Surveillance) | 37.50 | 54.87 | 45.70 | 35.18 | 23.03 | 29.30 | 37.14 | 74.76 | 55.95 |

### C.2.3. GENERALIZATION FROM THREE-CRITERIA TRAINING TO TWO-CRITERIA EVALUATION

To further evaluate the transferability of criterion adaptation across settings, we directly evaluate models trained under the three-criteria setting in the standard two-criteria scenario without additional fine-tuning. As shown in Table 12, several important observations emerge.

*Criterion adaptation learned under richer supervision transfers effectively.* Despite being trained under a richer three-criteria setting, MCT-trained models maintain strong adaptation performance under the two-criteria evaluation setting. Compared to the base model, substantial improvements in CS are consistently observed across both in-domain and cross-domain evaluations. For example, the model trained on the Industrial Inspection domain improves CS in Medical Diagnosis from 8.54% to 62.38%, while the model trained on the Video Surveillance domain improves CS in Video Surveillance from 0.00% to 34.29%. These results indicate that MCT does not depend on memorizing a fixed set of criterion-specific mappings, but instead learns a transferable mechanism for inferring decision boundaries from contextual examples.

*Cross-domain and reverse generalization remain robust across criterion settings.* The cross-domain transfer patterns observed in the standard two-criteria setting remain largely preserved after training under the richer three-criteria setting. Models trained on static-image domains continue to generalize effectively to other static domains, while transfer to Video Surveillance remains comparatively limited due to modality gaps. Conversely, the model trained on the Surveillance domain still demonstrates strong reverse generalization to static-image domains, achieving 43.53% CS on Industrial Inspection and 42.78% on Medical Diagnosis. This suggests that criterion adaptation learned from temporally complex video data remains transferable to simpler static-image scenarios across different criterion configurations.

Overall, these results provide further evidence that MCT induces a flexible and transferable criterion adaptation capability that generalizes across different criterion configurations, rather than merely memorizing discrete decision rules.

*Table 12.* Generalization from 3-criteria training to 2-criteria evaluation. Models tuned with MCT under the 3-criteria setting on a single domain are evaluated with 2-criteria on both the same and the remaining domains.

| Model | Industrial Inspection | | | Medical Diagnosis | | | Video Surveillance | | |
|---|---|---|---|---|---|---|---|---|---|
| | CS | CI | Ov. | CS | CI | Ov. | CS | CI | Ov. |
| *Qwen2.5-VL-7B-Instruct* | | | | | | | | | |
| Base | 13.79 | 39.74 | 27.70 | 8.54 | 31.35 | 20.09 | 0.00 | 28.57 | 14.29 |
| MCT (Industrial) | 62.93 | 77.43 | 70.70 | 62.38 | 40.86 | 51.48 | 13.33 | 33.33 | 23.33 |
| MCT (Medical) | 68.32 | 64.18 | 66.10 | 66.23 | 64.26 | 65.32 | 4.76 | 29.52 | 17.14 |
| MCT (Surveillance) | 43.53 | 66.23 | 55.70 | 42.78 | 40.86 | 41.81 | 34.29 | 83.81 | 59.05 |

## C.3. Additional Results of Effect of the Number of Shots on Criterion Adaptation

This appendix reports additional results of the context scaling ablation on the Medical Diagnosis and Video Surveillance domains. Following the same protocol as Ablation 3 in Section 5.5, base models are evaluated under 3-shot, 6-shot, and 9-shot settings. As shown in Figure 7, increasing the number of shots yields limited and unstable effects on CS across both domains, which is consistent with observations on Industrial Inspection.

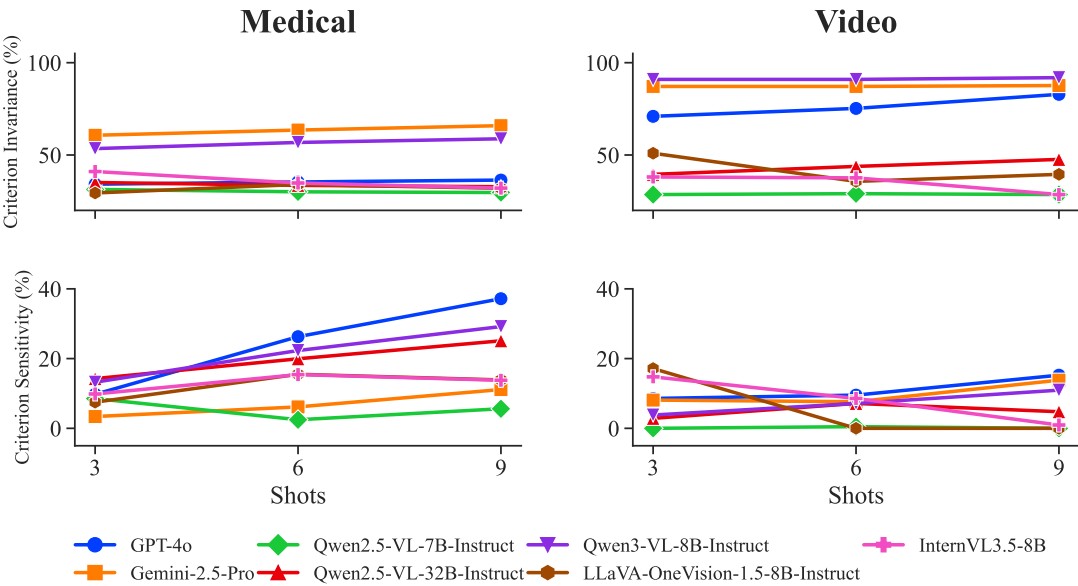

*Figure 7.* Effect of the number of shots on Medical Diagnosis and Video Surveillance. Similar to Industrial Inspection, increasing the number of shots yields limited improvements in Criterion-Sensitivity across both domains, particularly for open-source models.

## C.4. Analysis of Models' Response Distribution

To further analyze the behavioral manifestation of the rigid boundary bias explained in Section 5.2, we examine the joint distribution of model predictions under two alternative criteria. Figure 8 reports the distribution of prediction pairs across domains and models in the criterion-sensitive set.

Across all three domains, we observe that for most base models, over $80\%$ of samples receive identical predictions under both criteria (green and orange parts of pie charts), indicating that the decision boundary remains effectively unchanged despite explicit criterion shifts in the context. This phenomenon is especially pronounced for open-source models in Video Surveillance, where models such as Qwen2.5-VL-32B-Instruct and Qwen2.5-VL-7B-Instruct collapse entirely to a single prediction mode, yielding nearly $100\%$ identical responses and zero criterion-sensitive behavior.

As a complementary analysis to the aggregated CI/CS metrics, this prediction-pair distribution provides a more direct and intuitive view of model rigidity: rather than partially adapting, base models overwhelmingly default to a fixed task-level

*Table 13.* Comparison between different training data composition. Criterion-Specific (%), Criterion-Invariant (%), and Overall (%) metrics are reported across Industrial Inspection, Medical Diagnosis, and Video Surveillance. Base denotes the pretrained Qwen2.5-VL-7B-Instruct model without fine-tuning. Criterion A, Criterion B and MCT denote models trained using data from either criterion and two criteria together, respectively.

| Setting | Industrial Inspection | | | Medical Diagnosis | | | Video Surveillance | | |
|---|---|---|---|---|---|---|---|---|---|
| | CS | CI | Ov. | CS | CI | Ov. | CS | CI | Ov. |
| *Qwen2.5-VL-7B-Instruct* | | | | | | | | | |
| Base | 13.79 | 39.74 | 27.70 | 8.54 | 31.35 | 20.09 | 0.00 | 28.57 | 14.29 |
| Criterion A | 6.90 | 85.82 | 49.20 | 60.51 | 64.90 | 62.73 | 2.86 | 70.95 | 36.90 |
| Criterion B | 60.56 | 73.51 | 67.50 | 64.35 | 56.03 | 60.14 | 10.95 | 81.90 | 46.43 |
| MCT | 68.32 | 74.63 | 71.70 | 83.11 | 50.09 | 66.39 | 45.71 | 81.90 | 63.81 |

decision rule. These results corroborate our claim that standard VLMs primarily perform task recognition, while lacking the mechanism to dynamically reconfigure decision boundaries in response to criterion shifts.

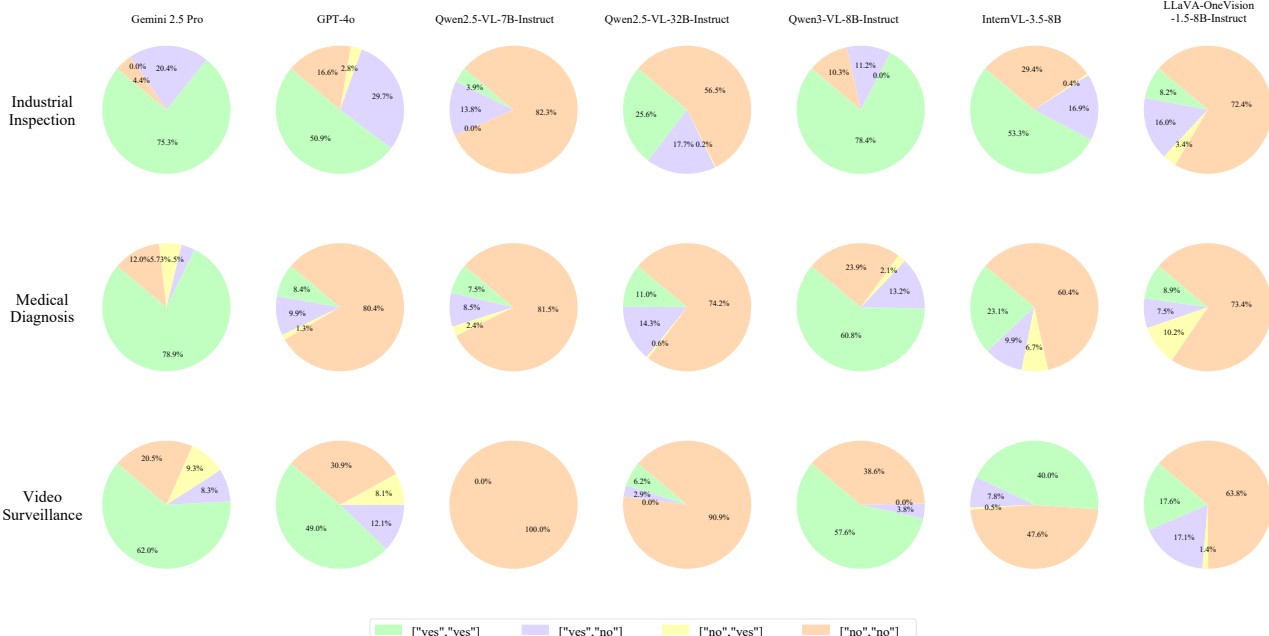

*Figure 8.* Statistical breakdown of model responses across two criteria. Each pie chart illustrates the percentage of response patterns under two criteria for seven large vision-language models across three domains: Industrial Inspection, Medical Diagnosis, and Video Surveillance. A dominant mass on identical prediction pairs (i.e., no label switch) indicates failure in criterion adaptation and reflects a rigid decision boundary. Colors for each pattern are as shown in the legend. For each pair in the legend, the former denotes the answer to Criterion A.

## C.5. Effect of Training Data Composition

Table 13 reports the performance of different training strategies across three CC-Bench domains. We conduct this ablation on Qwen2.5-VL-7B-Instruct.

We can observe that Criterion A does not consistently improve overall performance and, in some cases, exacerbates the discrepancy between CS and CI from under 30% to over 70%. This behavior suggests that Criterion A is closely aligned with the model's pretrained knowledge, causing fine-tuning to reinforce the existing rigid decision boundary rather than mitigate it.

Criterion B achieves more balanced CS and CI performance and yields more stable improvements across multiple domains. We attribute this trend to a larger distributional or semantic gap between Criterion B and the pretrained prior, which

encourages the model to relax its original boundary and partially mitigates rigid boundary bias.

MCT, trained jointly on data from both criteria, consistently achieves the best overall performance. By directly exposing the model to multiple criteria during training, MCT enables explicit modeling of criterion-conditioned variability, leading to strong CS performance across all domains. These results indicate that multi-criterion training is an effective approach for improving adaptation under the CC-ICL setting.

### C.6. Case Study

To illustrate the practical effect of MCT on VLMs, we present several representative cases from the CC-Bench dataset, covering Industrial Inspection, Medical Diagnosis, and Video Surveillance. Each case highlights a CS sample whose ground-truth label varies across different criteria, requiring the model to infer the latent decision criterion from the support set.

**Industrial Inspection.**   Figure 9 shows two inspection examples. The first case is from the *Bottle* category. The image contains contamination and structural damage, while the defect criterion is determined by the provided support set. Under Criterion A, surface contamination is sufficient to trigger an abnormal label, whereas Criterion B only considers severe structural damage as defective.

The second case (*Hazelnut*) corresponds to a crack-related defect that is consistently labeled as abnormal under both criteria. The base model fails to detect the defect under either criterion, as it relies on a rigid, criterion-agnostic decision boundary dominated by non-target visual cues.

Together, these cases show that MCT enables criterion-aware adaptation in Industrial Inspection.

**Medical Diagnosis.**   Figure 10 presents two medical imaging examples that share the same failure mode as the Industrial Inspection cases. In the lung CT case, the query targets *interstitial lung disease*, while the image also contains visual evidence of *bronchiectasis*, making the sample criterion-sensitive. In the brain MRI case, *white matter change* similarly corresponds to a non-queried criterion present in the image, making this example likewise criterion-sensitive.

Across both cases, the base model consistently fails by shifting its prediction toward non-queried criteria present in the image, whereas the model trained with MCT correctly follows the queried criterion and produces criterion-consistent predictions.

**Video Surveillance.**   Figure 11 presents a Video Surveillance example where the target action is specified by the query, while the decision criterion is implicitly defined by the support set. Consistent with the industrial and medical domains, the base model fails to adapt its prediction across different criteria, revealing a rigid boundary bias. After multi-criterion training, the model successfully aligns its judgment with the queried criterion and produces criterion-consistent predictions under varying video criteria.

**Discussion.**   These case studies illustrate a consistent failure mode of base VLMs. Across all domains, base models rely on fixed decision boundaries and fail to adjust their predictions when the underlying criterion changes, leading to errors on criterion-sensitive samples.

In contrast, models trained with multi-criterion data correctly infer the intended criterion from the support set and adapt their predictions accordingly. This behavior is consistent with the improvements observed in both CI and CS metrics.

Overall, these results indicate that the limitation of base models lies in criterion adaptation rather than visual understanding, and that multi-criterion training effectively mitigates rigid boundary bias across domains.

**Question**                    **Is there an anomaly in the image? Answer with yes/no.**

*Figure 9.* Industrial Inspection Case.

**Question**                          **Is this image abnormal? Answer with yes/no.**

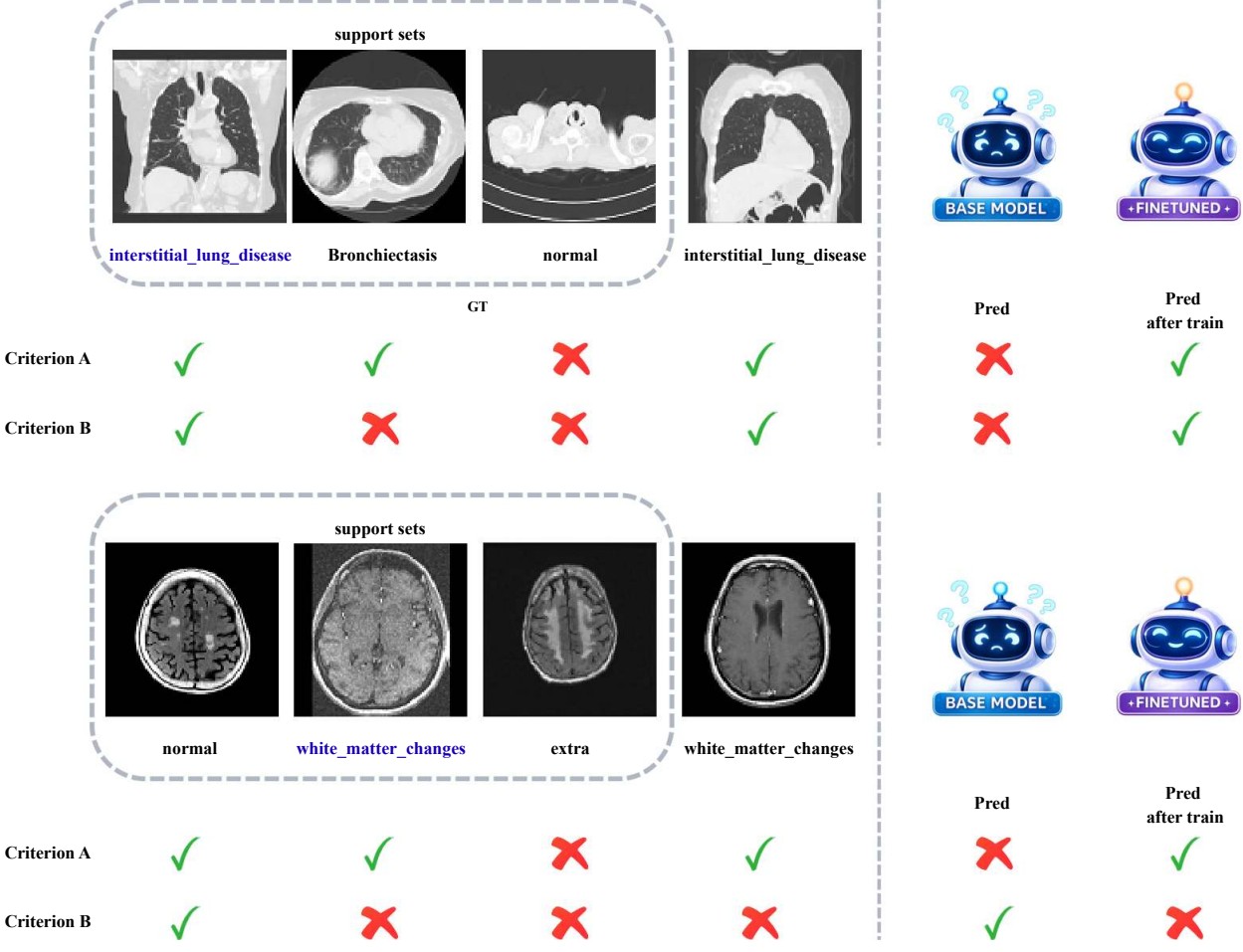

*Figure 10.* Medical Diagnosis Case.

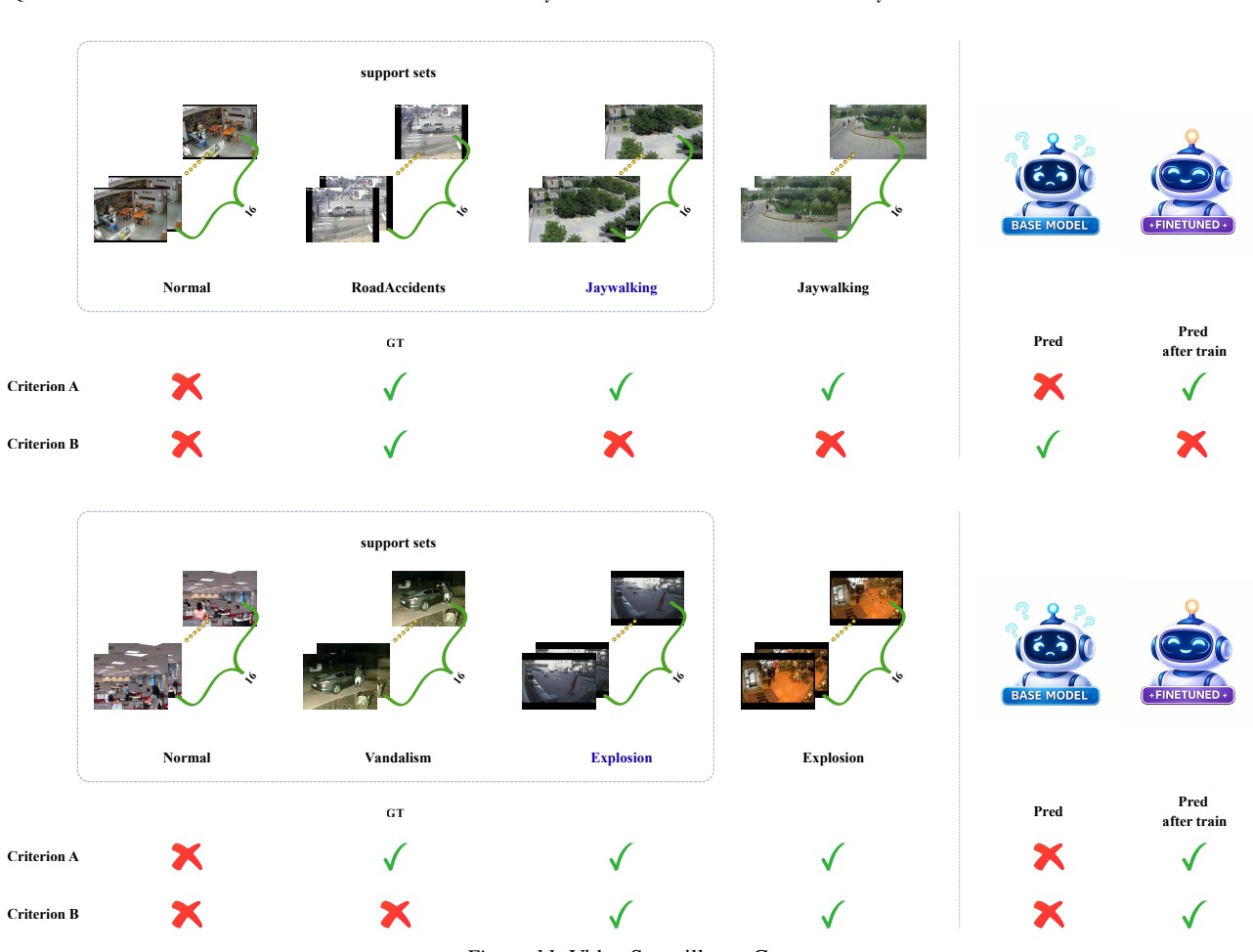

*Figure 11.* Video Surveillance Case.

