# OpenReview forum: "Criterion-Conditional In-Context Learning: Evaluating Criterion-Shift Adaptation in Vision-Language Models"
_ICML.cc/2026/Conference — ICML 2026 regular_

### Official Review · Reviewer_m14U · 2026-03-09

**Soundness:** 2
**Presentation:** 3
**Significance:** 2
**Originality:** 3
**Overall Recommendation:** 3
**Confidence:** 3

**Summary:**

The paper introduces Criterion-Conditional In-Context Learning (CC-ICL), a novel setting designed to evaluate whether Vision-Language Models (VLMs) can adapt their decision boundaries based on varying criteria demonstrated in the support set, while the high-level task remains fixed. To measure this, the authors propose two metrics: Criterion Invariance (CI) and Criterion Sensitivity (CS), and construct a multi-domain dataset, CC-Bench. The evaluation reveals that state-of-the-art VLMs suffer from a rigid boundary bias, defaulting to pre-trained task priors rather than adapting to contextual criteria. To address this, the authors propose Multi-Criterion Training (MCT), a fine-tuning approach that significantly improves criterion adaptation in 7B-parameter models.

**Compliance With Llm Reviewing Policy:**

Affirmed.

**Final Justification:**

The authors' response addressed my main concerns, so I have raised my score accordingly.

**Key Questions For Authors:**

1. How do you justify framing this work as a study of In-Context Learning when the proposed method (MCT) relies on explicit parameter updates through LoRA? If the underlying base models do not inherently possess the capability to adapt evaluation criteria via pure in-context mechanisms, does MCT simply demonstrate that the models can be instruction-tuned to respond to a specific prompt template, rather than providing evidence of an improved capacity for genuine in-context reasoning?

2. Your framework operationalizes criterion shifts by mapping nuanced subcategories into a binary “Normal/Abnormal” label. How would CC-ICL and the associated evaluation metrics generalize to open-ended VQA tasks or generative settings, where criterion shifts cannot be naturally reduced to binary label spaces?

**Limitations:**

yes

**Strengths And Weaknesses:**

**Strengths:**

* The paper addresses an important challenge in multimodal reasoning: the disentanglement of high-level task semantics from fine-grained, context-dependent decision criteria. This problem is of substantial practical significance, particularly in real-world deployment scenarios where decision thresholds must be adapted to varying operational requirements (e.g., different tolerance levels in industrial inspection tasks).

**Weaknesses:**

* The paper contains a fundamental methodological inconsistency. It explicitly frames the problem within the paradigm of In-Context Learning (ICL), which is inherently characterized by performing tasks without updating model parameters. However, the proposed MCT method relies on LoRA-based fine-tuning, which introduces supervised parameter updates. Addressing deficiencies in ICL through supervised fine-tuning fundamentally contradicts the defining premise of the ICL paradigm. Consequently, the proposed approach does not genuinely improve the model’s in-context learning capability; rather, it effectively teaches the model a new task format through instruction-style fine-tuning.

---

> ### Author Rebuttal · Authors · 2026-03-31
>
> We sincerely thank Reviewer m14U for the professional and insightful feedback, and for recognizing the practical significance of our work in real-world deployment scenarios. We address each concern in detail below.
>
> ### **Commitment to Open Release**
> Upon acceptance, we will publicly release our full codebase, trained checkpoints, and the whole dataset to ensure full reproducibility and support further community research.
>
> ---
>
> ### **Addressing Weaknesses and Key Questions**
> Below, we address the reviewer's concerns through key questions. Weakness 1 is discussed in KQ1-1 & KQ1-2.
>
> **KQ1-1: How do you justify framing this work as a study of In-Context Learning when the proposed method (MCT) relies on explicit parameter updates through LoRA?**
>
> We would like to clarify that ICL is defined as a test-time capability to **perform new tasks from demonstrations** without optimizing any parameters.
>
> Importantly, this definition **does not constrain how such ICL capability is acquired**. In practice, ICL is not a purely emergent ability, which often requires post-training to be effectively elicited or strengthened. Prior works (e.g. True-MICL[1] and SymDPO[2]) have already shown that SFT and RL training can enhance ICL ability. Recent work such as CL-bench[3] also argue that models must learn "context-aware" reasoning abilities during post-training to effectively acquire new knowledge from context.
>
> And we specifically design cross-domain experiments (Sec 5.4, Ablation 1) to verify that MCT equips the model with criterion adaptation capability, **rather than converting a new task into a learned one**.
>
> Therefore, MCT does not contradict the ICL definition.
>
> ---
>
> **KQ1-2: Does MCT simply demonstrate that the models can be instruction-tuned to respond to a specific prompt template, rather than providing evidence of an improved capacity for genuine in-context reasoning?**
>
> We think that the performance gains from MCT stem from an enhanced criterion adaptation capability rather than improved format compliance or template memorization.
>
> **1. Failure of Base Models is Not Due to Format**: We found that base models already follow the yes/no answer format with almost 100% accuracy as shown in the table below, yet still exhibit CS lower than 20%. This indicates that their failures stem from an inability to adapt to criteria, rather than formatting issues.
>
> |Model|Avg. Format Acc|
> |-|-|
> |GPT-4o|90.2|
> |Gemini 2.5 Pro|98.7|
> |Qwen2.5-VL-7B-Instruct|100|
> |Qwen2.5-VL-32B-Instruct|100|
> |Qwen3-VL-8B-Instruct|100|
> |InternVL3.5-8B|99.9|
> |LLaVA-OneVision-1.5-8B-Instruct|100|
>
> **2. Cross-Domain Generalization Demonstrates Transferable Criterion Adaptation**: If MCT only provided template-specific tuning, its effects would be confined to the training distribution. Crucially, our cross-domain experiments (Sec 5.4, Table 3) demonstrate that a model trained with MCT on one specific domain generalizes its criterion adaptation ability to:
>   - **Unseen Domains**: Generalization across 3 domains.
>   - **Diverse Instructions**: Handling diverse instruction phrasings (Appendix B, Table 8).
>   - **Cross-Modality**: Transferring from video to image modality.
>
> The fact that MCT-trained models consistently outperform baselines across these diverse settings, proves that the model has learned a meta-level reasoning strategy: inferring the latent criterion from the support set and conditioning the prediction accordingly.
>
> ---
>
> **KQ2: Generalization beyond Binary Questions**
>
> The whole evaluation framework is not tied to any specific label space and extends to tasks with varying outputs, as long as multiple valid evaluation criteria exist.
>
> **1. Formulation Perspective**
>
> As formalized in Sec. 3.1, CC-ICL specifically targets tasks with multiple valid mappings (i.e., $∣M∣>1$ ) under fixed task semantics. This formulation is agnostic to the definition of input and output spaces.
>
> **2. Metrics Perspective**
>
> As defined in Sec. 3.2, CI and CS measure whether a model can produce correct answers under different criterion-specific context for a given query. Importantly, they do not impose any restriction on how "correctness" is defined, which can be evaluated using any task-dependent evaluation schemes.
>
> For open-ended VQA or generative tasks, criterion shifts can be instantiated via different focus preferences (e.g., atmosphere-focused vs. primary-object-focused) and LLM-as-a-judge can be used as an evaluation method, enabling CI/CS computation beyond discrete labels.
>
> We view CC-Bench as a controlled testbed separating invariant and sensitive samples, and CC-ICL as a general paradigm for evaluating criterion-conditioned reasoning beyond it.
>
> ---
> **References**
>
> [1] True Multimodal In-Context Learning Needs Attention to the Visual Context
>
> [2] SymDPO: Boosting In-Context Learning of Large Multimodal Models with Symbol Demonstration Direct Preference Optimization
>
> [3] CL-bench: A Benchmark for Context Learning

---

> > ### Author Rebuttal · Reviewer_m14U · 2026-04-02
> >
> > Thank you for your detailed response. I have raised my score accordingly.

---

> > > ### Author Response · Authors · 2026-04-06
> > >
> > > We sincerely thank Reviewer m14U for the increased score and for confirming that all concerns have been fully resolved. The questions raised during the first round, particularly regarding the relationship between MCT and the ICL paradigm, were sharp and constructive, and helped us articulate the positioning of our work more clearly.
> > >
> > > We would also like to highlight **additional experiments** conducted during the rebuttal period that **further strengthen the paper's conclusions with regard to KQ1-2**:
> > >
> > > ---
> > > ### **Multi-criteria Generalization**
> > > Models trained under two criteria are evaluated under a three-criteria setting. The results (see table below) show that CS does not degrade significantly compared to Table 3, confirming that MCT teaches the model to adapt to shifted decision boundaries rather than memorizing a fixed set of criteria. Furthermore, training and testing under the three-criteria setting improves CS by 40% over the baseline, while maintaining strong cross-domain and reverse generalization (train on three, test on two). Full results are provided in link*.
> > >
> > > |model|Industrial Inspection(CS/CI/Ov.)|Medical Diagnosis(CS/CI/Ov.)|Video Surveillance(CS/CI/Ov.)|
> > > |-|-|-|-|
> > > |Qwen2.5-VL-7B-Instruct|12.12/44.07/27.20|3.75/29.34/16.71|0.00/28.58/14.29|
> > > |MCT (Industrial)|55.60/66.79/61.60|50.28/33.73/41.90|10.95/31.43/21.19|
> > > |MCT (Medical)|45.47/48.32/47.00|63.98/32.18/47.87|0.95/28.57/14.76|
> > > |MCT (Surveillance)|41.59/54.66/48.60|29.92/23.58/26.71|42.38/66.67/54.52|
> > >
> > > ---
> > > ### **Chain-of-Thought Analysis**
> > > By prompting models to generate reasoning traces alongside predictions, we observe a clear qualitative distinction: the **base model** reasons purely from the query image **without any reference to the support set**, whereas the **MCT-trained model** explicitly performs visual comparisons between the query and each support image, **grounding its prediction in the visually demonstrated criterion** rather than abstract textual patterns. A representative example from the industrial inspection domain is shown below:
> > >   - **Base model:** "Shape: The nut appears to be roughly spherical... Surface Texture: The surface has a natural, slightly rough texture... Given these observations, there does not appear to be any anomaly in the image... Final answer: No."
> > >   - **MCT-trained model:** "Compare the image to the first image: The first image shows a hazelnut with a small crack on its surface. Compare the image to the second image: The second image shows a hazelnut without any visible cracks... The fourth image has a similar anomaly to the third example. Therefore, the fourth image also has an anomaly."
> > >
> > > This directly demonstrates that MCT elicits criterion-conditioned visual reasoning rather than criterion-agnostic prediction mapping. We show detailed cases of the base model and the trained model in link*.
> > >
> > > ---
> > > These results further consolidate the evidence that **MCT elicits genuine criterion adaptation rather than memorization of fixed criteria or input-output patterns**.
> > >
> > > We believe that the primary and lasting contribution of this work lies in the formalization of the CC-ICL setting and the construction of CC-Bench, which together provide the community with a new lens for evaluating criterion-conditioned reasoning in VLMs. MCT serves as an initial validation that this capability gap can be effectively addressed, and we hope this work motivates broader exploration in this direction.
> > >
> > > ---
> > > *: https://anonymous.4open.science/r/Criterion-Conditional-In-Context-Learning/README.md

---

### Official Review · Reviewer_ByDF · 2026-03-12

**Soundness:** 3
**Presentation:** 2
**Significance:** 3
**Originality:** 3
**Overall Recommendation:** 4
**Confidence:** 3

**Summary:**

The authors propose a new setting for multimodal in-context learning (MM ICL) that relies on multiple underlying mappings between inputs and outputs, which they call decision criteria. Specifically, the authors consider a criterion adaptation beyond task induction and propose two evaluation metrics, criterion invariance and criterion sensitivity, to assess the robustness of MM ICL under dynamic and shifting criteria. Then, they propose a benchmark of dynamic criteria by extending real-world tasks and LoRA fine-tuning techniques to improve a model's criterion-adaptation capabilities. The experimental results demonstrate the promising performance of LoRA fine-tuned models on the benchmark.

**Compliance With Llm Reviewing Policy:**

Affirmed.

**Final Justification:**

Please see my Rebuttal Acknowledgment (https://openreview.net/forum?id=yGEllPWlnM&noteId=VLAeai2QI9).

In summary,
I like the paper's direction in proposing this approach to investigate the MM ICL, and the author introduces a new real-world benchmark for evaluating MM ICL under different decision boundaries, which is good for this research direction.

> Reviewer UN9i
> By conceding that the core metrics (CI and CS) ignore underlying cross-modal reasoning, it fails to prove whether the massive MCT gains stem from genuine criterion-adaptation or learning to change predictions without reasoning.

I agree with UN9i's viewpoints on the limitations of the current evaluation metrics. If the authors could design the metric to clarify which factors affect the predictions, that would be better.

**Key Questions For Authors:**

1. When establishing **CC-Bench**, the authors leverage GPT-4o to perform the ranking of sub-categories as an initial rank. This process implies that GPT-4o is sufficiently accurate on these datasets, which raises my concerns about their reproducibility (the stability of GPT-4o's output) and the process's generalizability to other datasets (which requires domain experts).
2. The process of MCT is unclear to me. In concrete terms, how do you fine-tune a model on the binary class from a shared triplet of instances?
	1. Are tasks in **CC-Bench** designed as binary classification tasks? (convert the hierarchical design of **Category** and **Sub-category** to "Normal" and "Abnormal")
	2. How does the criterion information (Criterion A, Criterion B) reveal to a model when fine-tuning?
3. The current experimental design relies on a comparison between 'zero-tolerance' and 'limited tolerance' criteria for studying the criterion-shift adaptation. To strengthen the paper, the authors could evaluate the 'medium-tolerance' criterion, which would provide a more nuanced understanding of decision boundaries than the two criteria do.
4. **Rigid boundary bias.** The authors attribute the low CS (Criterion Sensitivity) scores exclusively to "rigid boundary bias" (i.e., the model's inability to shift its decision boundary). However, it remains unclear whether the failure stems from a lack of *visual grounding* or a failure to follow context. For example, could you provide a model's descriptions of the "Abnormal" case to check the reasons for the failure prediction?

**Limitations:**

1. I consider that the real-world case might involve several underlying factors that affect a model's performance (domain knowledge, image quality, prompting, etc.). I suggest that the authors can provide a simpler case to study the criterion-shift adaptation. For example, the number of 'outlier' shapes as criterions in the Outlier Detection task of TrueMICL.

**Strengths And Weaknesses:**

*Soundness*

1. The proposed framework looks reasonable.
2. However, I have several concerns about the proposed datasets, the multi-criterion training (MCT) process, and the terms of *Rigid Boundary Bias* in analysis.

*Presentation*

3. The motivation of this work is clear.
4. However, the process of MCT and its success in overcoming the static decision boundary are unclear to me.

*Significance*

5. The authors provide the different aspects of MM ICL, which departs from investigating the *visual negligence* in the previous benchmark. Overall, I like the paper's direction in proposing this approach to investigate the MM ICL.

*Originality*

6. This work introduces the new real-world benchmark for evaluating MM ICL under different decision boundaries.

---

> ### Author Rebuttal · Authors · 2026-03-31
>
> We sincerely thank Reviewer ByDF for the insightful and constructive review. We are very grateful for the recognition of the paper’s unique perspective in investigating multimodal in-context learning, and the introduction of the new real-world benchmark. We address each concern in detail below.
> ### **Commitment to Open Release**
> Upon acceptance, we will publicly release our full codebase, trained checkpoints, and the whole dataset to ensure full reproducibility and support further community research.
>
> ---
>
> ### **Addressing Weaknesses and Key Questions**
> **KQ1: Explanation on the Role of GPT-4o in Dataset Construction and its Impact on Reproducibility and Generalizability**
>
> As described in Sec 4.3, GPT-4o serves only as an optional tool which can be replaced by standard human annotation process, and does not affect the reproducibility or generalizability of the data construction pipeline.
>
> **Regarding reproducibility**, our pipeline does not depend on GPT-4o: it is used only for an initial ranking of sub-categories, which is subsequently refined by domain experts. The final CC-Bench data is therefore reliable.
>
> **Regarding generalizability**, introducing new domains only requires defining an ordering over sub-categories; for domains where SoTA models perform poorly, this can be directly specified by domain experts.
>
> ---
>
> **KQ2: Clarification on MCT and CC-Bench Formulation**
>
> **(1) Are tasks in CC-Bench designed as binary classification tasks?**
>
> Yes, but we'd like to emphasize that the CC-ICL setting and metrics are both inherently task-agnostic and generalizable beyond binary questions (as detailed in Sec. 3).
>
> **(2) How do you fine-tune a model on the binary class from a shared triplet of instances?**
>
> For each shared triplet $s_{\text{abn}}, s_{\text{norm}}, s_{\text{sen}}$ , we build a pair of support sets $S_A, S_B$ that contains identical images but differs only in the label of the criterion-sensitive instance $s_{\text{sen}}$. Training instances are constructed by combining the same query with each support set and randomly sampled during training.
>
> As a result, during training, the model observes:
> - the same query and same visual inputs,
> - different answers conditioned on different support sets (criteria).
>
> This ensures that supervision is criterion-conditioned rather than label-static.
>
> We provide the complete template and cases in Appendix B.2, C.4.
>
> **(3) How is criterion information revealed to the model?**
>
> Criterion information is not explicitly provided (e.g., no "Criterion A/B" special tokens or direct instructions).
>
> Instead, it is implicitly encoded in the support set, requiring the model to infer and adapt to the latent criterion from context.
>
> As a result, model's criterion adaptation ability arises from in-context reasoning over the support set, rather than direct criterion signals.
>
> ---
>
> **KQ3: Evaluation of Multiple Criteria**
>
> We agree that introducing more criteria can provide a more fine-grained evaluation of model behavior under criterion shifts. For simplicity, we extend the two-criteria setup by adding a third one.
>
> When evaluating models trained on two criteria under this three-criteria setting, CS does not degrade significantly compared to Table 3, suggesting that models can adapt to shifted decision boundaries rather than memorizing a fixed number of criteria.
>
> |model|Industrial Inspection(CS/CI/Ov.)|Medical Diagnosis(CS/CI/Ov.)|Video Surveillance(CS/CI/Ov.)|
> |-|-|-|-|
> |Qwen2.5-VL-7B-Instruct|12.12/44.07/27.20|3.75/29.34/16.71|0.00/28.58/14.29|
> |MCT(Industrial)|55.60/66.79/61.60|50.28/33.73/41.90|10.95/31.43/21.19|
> |MCT(Medical)|45.47/48.32/47.00|63.98/32.18/47.87|0.95/28.57/14.76|
> |MCT(Surveillance)|41.59/54.66/48.60|29.92/23.58/26.71|42.38/66.67/54.52|
>
> Furthermore, train and test under the three-criteria setting improves CS by 40% over the baseline, while maintaining strong cross-domain and reverse generalization (train on three, test on two). Full results are provided in link*.
>
> ---
>
> **KQ4: Reasons for Failure Prediction**
>
> We prompt models to explicitly explain their reasoning when predicting answers.
> - **Base models** tend to reason solely on query image features, without referencing the support set.
> - **MCT-trained models** condition their reasoning on the support set.
>
> We show detailed cases of the base model and the trained model in link*.
>
> ---
>
> **L1: Simpler Case for Underlying Factors Analysis**
>
> We agree that although simplified settings ca help isolate specific factors, they may not fully reflect the complexity of criterion adaptation in practice. CC-Bench is designed with the motivation to evaluate models under real-world scenarios, while still maintaining controllability through structured design.
>
> We will include as future work the development of more controlled and fine-grained benchmarks to better analyze criterion adaptation at a more isolated level.
>
> *: https://anonymous.4open.science/r/Criterion-Conditional-In-Context-Learning/README.md

---

> > ### Author Rebuttal · Reviewer_ByDF · 2026-04-04
> >
> > Thank the authors for the detailed response. All my questions are resolved.
> > As the MM ICL practitioner, I am looking forward to the release of this benchmark.
> > I will increase my score.
> >
> > Last thing I want to highlight to the AC and the authors: I am unsure whether providing a link to an external resource during the rebuttal process is allowed or fair to other submissions.

---

> > > ### Author Response · Authors · 2026-04-06
> > >
> > > We sincerely thank Reviewer ByDF for the increased score and for the recognition of our CC-ICL setting and CC-Bench. The detailed questions raised during the first round were very helpful in strengthening the paper. In the camera-ready version, we will incorporate the clarifications on MCT, the visualization analysis of base model failures vs. MCT successes, and the multi-criteria experiments. The full codebase, dataset, and trained checkpoints will also be publicly released.
> > >
> > > Regarding the point about external links: according to the official ICML rebuttal notifications, anonymous links are permitted, with the constraint that "links may only be used for figures (including tables) and captions that describe the figure (no additional text)." Our usage fully complies with these guidelines. We appreciate the reviewer for flagging this concern.

---

### Official Review · Reviewer_UN9i · 2026-03-12

**Soundness:** 2
**Presentation:** 2
**Significance:** 3
**Originality:** 3
**Overall Recommendation:** 3
**Confidence:** 2

**Summary:**

This paper studies an evaluation setting called Criterion-Conditional In-Context Learning (CC-ICL) to assess how well VLMs adjust their decision boundaries under shifting criteria. Two complementary metrics Criterion Invariance and Criterion Sensitivity are proposed to capture the model’s robustness and adaptability under criterion shifts. Meanwhile, a multi-domain benchmark CCBench is constructed to support evaluation under the CC-ICL setting. CC-Bench enables legitimate ground-truth variation conditioned on the active criterion.

**Compliance With Llm Reviewing Policy:**

Affirmed.

**Final Justification:**

I appreciate the authors' detailed responses. Nevertheless, the provided clarification does not fundamentally alter my assessment of the weaknesses. Consequently, my initial score remains unchanged.

**Key Questions For Authors:**

-- Experimental results in Table 1 reveal a severe "rigid boundary bias" in VLMs, where they seem to rely on static task-level priors rather than dynamic criterion adaptation. For instance, Qwen2.5-VL-7B-Instruct achieves a 0.00% CS score  in the video domain, failing to adjust decision boundaries upon context.
-- Experiment results demonstrate that models trained on static image domain (Medical) fail to generalize to dynamic video surveillance tasks. It seems to be unfit for models trained on static images to obtain the temporal induction ability to temporal context.
-- Increasing the number of shots (from 3 to 9) yields inconsistent improvement in CS score, particularly for open-source models. This result suggests that rigid boundary bias cannot be resolved via simple prompt engineering or context scaling.
-- CI and CS metrics adopt simple weighting. These metrics do not explicitly consider semantic dependencies between modalities, which may introduce evaluation biases in complex samples where critical semantic details do not align with token.

**Limitations:**

yes

**Strengths And Weaknesses:**

Strengths:
-- It introduces Criterion-Conditional In-Context Learning (CC-ICL) to address how models adapt to shifting decision.
-- The authors designe two specialized metrics: Criterion Invariance (CI) and Criterion Sensitivity (CS). CI measures robustness by ensuring consistent predictions for samples unaffected by the shift. CS quantifies adaptability with a specific criterion.
-- The constructed CC-Bench provides a robust testing ground across different domains. It utilizes a dual-level data hierarchy to enable legitimate ground-truth variation conditioned on the active criterion even when the task remains fixed.

Weaknesses:
-- Experimental results in Table 1 reveal a severe "rigid boundary bias" in VLMs, where they seem to rely on static task-level priors rather than dynamic criterion adaptation. For instance, Qwen2.5-VL-7B-Instruct achieves a 0.00% CS score  in the video domain, failing to adjust decision boundaries upon context.
-- Experiment results demonstrate that models trained on static image domain (Medical) fail to generalize to dynamic video surveillance tasks. It seems to be unfit for models trained on static images to obtain the temporal induction ability to temporal context.
-- Increasing the number of shots (from 3 to 9) yields inconsistent improvement in CS score, particularly for open-source models. This result suggests that rigid boundary bias cannot be resolved via simple prompt engineering or context scaling.
-- CI and CS metrics adopt simple weighting. These metrics do not explicitly consider semantic dependencies between modalities, which may introduce evaluation biases in complex samples where critical semantic details do not align with token.

---

> ### Author Rebuttal · Authors · 2026-03-30
>
> We are deeply grateful to Reviewer UN9i for the constructive review. We highly appreciate the reviewer's recognition of our proposed CC-ICL design and the robust testing dataset provided by CC-Bench across various domains. In the following sections, we address the specific concerns in detail.
> ### **Commitment to Open Release**
> Upon acceptance, we will publicly release our full codebase, trained checkpoints, and the whole dataset to ensure full reproducibility and support further community research.
>
> ---
>
> ### **Addressing Weaknesses and Key Questions**
> We would like to clarify that key questions 1–3 are in fact core empirical findings already reported and analyzed in our paper, rather than weaknesses overlooked by our work.
>
> **KQ1: Rigid Boundary Bias** is a central finding and motivation of our study. As shown in Table 1, most VLMs exhibit a severe imbalance between CI and CS, indicating reliance on static task-level priors instead of criterion adaptation. The extreme case (e.g., 0.00% CS in video) is explicitly discussed as evidence of failure in criterion-conditioned decision boundary adjustment.
>
> **KQ2: Limited Transferability (Image to Video)** is also analyzed in our cross-domain experiments in Sec. 5.4  (Ablation 1). We explicitly show that models trained on static domains struggle on video surveillance due to lack of temporal induction ability, highlighting a modality gap that is worth further exploration in future work.
>
> **KQ3: Limited Gains from Increasing Shots** is a reported result in Sec. 5.4 (Ablation 2). Our experiments demonstrate that simply scaling context does not reliably improve CS, especially for open-source models, supporting our conclusion that prompt engineering alone cannot resolve rigid boundary bias, which motivates us to design MCT.
>
> Therefore, these observations are intentional diagnostic results enabled by CC-Bench, designed to reveal the limitations of current VLMs under criterion shifts.
>
> ---
>
> **KQ4: Regarding Metrics Design**
>
> We thank the reviewer for this insightful comment. CI and CS are designed to focus on whether predictions appropriately change under criterion shifts, rather than modeling the underlying cross-modal reasoning process.
> We consider fine-grained evaluation an important direction for future work.

---

> > ### Author Rebuttal · Reviewer_UN9i · 2026-04-04
> >
> > I appreciate the additional clarifications in the rebuttal, which resolve most of my questions regarding the empirical finding, but they do not sufficiently change my overall assessment of the paper's weaknesses, so I keep my original score unchanged.
> >
> > The paper would still benefit from a clearer discussion of its evaluation limitations. By conceding that the core metrics (CI and CS) ignore underlying cross-modal reasoning, it fails to prove whether the massive MCT gains stem from genuine criterion-adaptation or learning to change predictions without reasoning.

---

> > > ### Author Response · Authors · 2026-04-06
> > >
> > > We thank Reviewer UN9i for the continued and thoughtful engagement. We fully acknowledge that CI and CS, as outcome-based metrics, cannot directly verify the internal reasoning process underlying model predictions. Therefore we provide three complementary lines of evidence below to address whether MCT gains stem from genuine criterion adaptation or from learning to change predictions without reasoning.
> > >
> > > ---
> > > **Qualitative evidence from reasoning traces.**
> > >
> > > The most direct evidence comes from chain-of-thought analysis. A representative example is shown below:
> > >
> > > 1. **Base model:** "Shape: The nut appears to be roughly spherical... Surface Texture: The surface has a natural, slightly rough texture... Given these observations, there does not appear to be any anomaly in the image... Final answer: No."
> > > 2. **MCT-trained model:** "Compare the image to the first image: The first image shows a hazelnut with a small crack on its surface. Compare the image to the second image: The second image shows a hazelnut without any visible cracks... The fourth image has a similar anomaly to the third example. Therefore, the fourth image also has an anomaly."
> > >
> > > The **base model** reasons purely from the query image **without any reference to the support set**, whereas the **MCT-trained model** explicitly performs visual comparisons between the query and each support image, **grounding its prediction in the visually demonstrated criterion** rather than abstract textual patterns. This directly demonstrates that MCT elicits criterion-conditioned visual reasoning rather than criterion-agnostic prediction mapping. More details are provided in the anonymous link*.
> > >
> > > ---
> > > **Quantitative evidence of genuine criterion adaptation from MCT.**
> > >
> > > Beyond the qualitative evidence above, two sets of experiments provide strong evidence against the hypothesis of prediction change without reasoning:
> > >
> > > 1. **cross-domain generalization** (Table 3): MCT-trained models consistently improve CS on unseen domains that differ in visual content, instruction phrasing, and even modality. If MCT merely taught the model to memorize domain-specific templates, such broad transfer would not be expected.
> > > 2. **multi-criteria generalization** (provided in the first-round rebuttal): models trained under two criteria successfully adapt to a three-criteria setting at test time (as shown below), requiring the model to handle unseen combinations of criteria rather than relying on fixed training configurations. Such generalization cannot be achieved by memorizing support-set label patterns or simple heuristics (e.g., majority voting via textual labels in the support set), which are inherently tied to specific training configurations and would not transfer to unseen criterion combinations.
> > >
> > > |model|Industrial Inspection(CS/CI/Ov.)|Medical Diagnosis(CS/CI/Ov.)|Video Surveillance(CS/CI/Ov.)|
> > > |-|-|-|-|
> > > |Qwen2.5-VL-7B-Instruct|12.12/44.07/27.20|3.75/29.34/16.71|0.00/28.58/14.29|
> > > |MCT (Industrial)|55.60/66.79/61.60|50.28/33.73/41.90|10.95/31.43/21.19|
> > > |MCT (Medical)|45.47/48.32/47.00|63.98/32.18/47.87|0.95/28.57/14.76|
> > > |MCT (Surveillance)|41.59/54.66/48.60|29.92/23.58/26.71|42.38/66.67/54.52|
> > >
> > > ---
> > > **On the evaluation design of CI/CS metrics.**
> > >
> > > We agree with the reviewer that CI and CS as outcome-based metrics do not directly model the underlying reasoning process. However, compared with standard accuracy metrics, our evaluation incorporates several deliberate design choices that provide substantially stronger guarantees for validating criterion adaptation:
> > >
> > > 1. The paired design of CI and CS enforces criterion invariance and sensitivity, where **CI prevents shortcut strategies such as majority voting over support labels, and CS ensures correct prediction changes under shifted criteria**, requiring the model to align visual inputs with the support set rather than relying on text-only patterns.
> > > 2. The support sets are randomly shuffled during training and evaluation, preventing the model from exploiting positional shortcuts such as always copying the label at a fixed position.
> > > 3. The test set is constructed with an approximate 1:1 ratio between criterion-invariant and criterion-sensitive samples, ensuring that a model strong on only one of CI or CS cannot achieve a high overall score.
> > >
> > > ---
> > > In summary, these three complementary lines of evidence, from reasoning trace analysis, generalization experiments, and evaluation design, collectively demonstrate that **MCT improvements stem from genuine criterion adaptation rather than superficial prediction changes without underlying reasoning**.
> > >
> > > We agree with the reviewer that developing process-level metrics is an important open problem, and will incorporate a clearer discussion of CI/CS limitations in the revised manuscript, explicitly noting that outcome-based metrics cannot fully rule out sophisticated pattern matching as an alternative explanation.
> > >
> > > ---
> > > *: https://anonymous.4open.science/r/Criterion-Conditional-In-Context-Learning/README.md

---

### Official Review · Reviewer_scc8 · 2026-03-23

**Soundness:** 3
**Presentation:** 3
**Significance:** 3
**Originality:** 3
**Overall Recommendation:** 4
**Confidence:** 3

**Summary:**

This study introduces CC-ICL and CC-Bench to assess VLM adaptability to shifting decision standards. Addressing a prevalent "rigid boundary bias," the authors propose Multi-Criterion Training (MCT), enabling 7B models to dynamically calibrate decision boundaries and rival proprietary systems in sensitivity.

**Compliance With Llm Reviewing Policy:**

Affirmed.

**Final Justification:**

I thank the authors for their detailed rebuttal. The additional cross-domain evidence and the three-criteria experiment effectively address my concerns regarding the generalization of MCT beyond simple binary memorization. I will maintain my Weak Accept recommendation.

**Key Questions For Authors:**

Can the CC-Bench framework and metrics be extended beyond binary anomaly detection to more complex or subjective reasoning tasks, such as open-ended visual description or multi-class judgment?

**Limitations:**

Please see the Key Questions box.

**Strengths And Weaknesses:**

**Strengths**

- Well-Motivated Problem Identification: The study identifies a critical "rigid boundary bias" in current VLMs, highlighting their struggle to recalibrate decision boundaries even when task semantics remain fixed.
- High-Quality CC-Bench Framework: It introduces a robust, multi-domain benchmark featuring a dual-level data hierarchy that enables the systematic evaluation of criterion-shift adaptation across industrial, medical, and surveillance tasks.
- Exceptional Performance Gains: The proposed MCT strategy allows 7B-scale open-source models to reach or even exceed the criterion sensitivity of proprietary frontier models like GPT-4o.

**Weaknesses**

- Heavy Dependency on Fine-Tuning: The performance improvements rely significantly on Multi-Criterion Training (MCT), suggesting that models still lack a true, parameter-free zero-shot capability for criterion adaptation.
- Limited Task Diversity: The evaluation is primarily focused on anomaly and defect detection tasks, which restricts the demonstration of the framework's applicability to more diverse or subjective reasoning scenarios.

---

> ### Author Rebuttal · Authors · 2026-03-30
>
> We sincerely thank Reviewer scc8 for the valuable and constructive feedback. We are grateful for the recognition of our core contributions in identifying the "rigid boundary bias" of VLMs and the high-quality CC-Bench dataset, as well as the exceptional performance gains achieved by our MCT strategy. We address each concern in detail below.
> ### **Commitment to Open Release**
> Upon acceptance, we will publicly release our full codebase, trained model weights, and the whole dataset to ensure full reproducibility and support further community research.
>
> ---
>
> ### **Addressing Weaknesses and Key Questions**
> **W1: Concerns about Fine-Tuning**
>
> **Before MCT, we have tried several training-free ways to improve criterion adaptation.** As shown in Sec. 5.4 (Ablation 2 & 3), we evaluate (i) increasing the number of shots and (ii) pure text-based prompting that explicitly describes the criterion. However, both approaches fail to yield improvements in Criterion Sensitivity (CS). Increasing shots leads to only marginal and inconsistent gains, while text-only prompts even reinforce rigid decision boundaries. These results demonstrate that criterion adaptation in the CC-ICL setting **cannot be obtained via context engineering alone**, and requires criterion-aware training like MCT.
>
> **(1) How MCT works.**
>
> MCT trains on paired contexts with identical set of images but different criterion-dependent labels. This prevents memorizing fixed mappings and forces the model to infer the active criterion from the support set and adjust its decision boundary accordingly.
>
> **(2) Cross-domain evidence.**
>
> If MCT only learned supervised mappings, gains would not transfer. However, as shown in Sec. 5.4 (Ablation 1), we observe consistent improvements in CS on entirely unseen domains. This shows that **MCT learns a true domain-agnostic adaptation ability rather than memorizing task-specific patterns or even converting a new task into a learned one**.
>
> **(3) Multi-criteria evidence.**
>
> We extend the two-criteria setup by adding a third one.
>
> When evaluating models trained on two criteria under this three-criteria setting, CS does not degrade significantly compared to Table 3, suggesting that models can truely adapt to shifted decision boundaries rather than memorizing a fixed number of criteria.
>
> |model|Industrial Inspection(CS/CI/Ov.)|Medical Diagnosis(CS/CI/Ov.)|Video Surveillance(CS/CI/Ov.)|
> |-|-|-|-|
> |Qwen2.5-VL-7B-Instruct|12.12/44.07/27.20|3.75/29.34/16.71|0.00/28.58/14.29|
> |MCT(Industrial)|55.60/66.79/61.60|50.28/33.73/41.90|10.95/31.43/21.19|
> |MCT(Medical)|45.47/48.32/47.00|63.98/32.18/47.87|0.95/28.57/14.76|
> |MCT(Surveillance)|41.59/54.66/48.60|29.92/23.58/26.71|42.38/66.67/54.52|
>
> Overall, we think MCT is an intuitive and effective way to evoke criterion adaptation capability.
>
> ---
>
> **W2 & KQ1: Concerns about Task Diversity, Framework and Metrics Generalizability**
>
> **The Diversity of CC-Bench**: We emphasize that CC-Bench is not restricted to a single task type, but is designed as a multi-domain benchmark with diverse semantics, including: Industrial inspection (low-level visual defects), Medical diagnosis (high-level semantic reasoning) and Video surveillance (temporal and behavioral understanding). **These domains differ substantially in modality, captured perspective, and reasoning requirements.**
>
> **The Generalizability of Framework and Metrics**: Our CC-ICL setting and metrics are both inherently task-agnostic and generalizable beyond binary questions.
>   1. **Generalizability of the CC-ICL**: As formalized in Sec. 3.1, CC-ICL specifically targets tasks with multiple valid mappings (i.e., $∣M∣>1$ ) under fixed task semantics. This formulation is agnostic to the definition of input and output spaces, and therefore naturally generalizes beyond binary classification to tasks like preference-based ranking, multi-class prediction, visual description and broader criterion-dependent decision-making scenarios.  For example, in open-ended visual description, criterion shifts can be instantiated via different output requirements (e.g., atmosphere-focused vs. primary-object-focused).
>   2. **Generalizability of CI and CS Metrics**: As defined in Sec. 3.2, the proposed metrics (CI and CS) only consider whether predictions are correct across all criteria. It does not constrain how correctness is defined, and can be readily applied to any tasks that require either rule-based or model-based evaluations with diverse output spaces. For example, in open-ended settings, correctness can be instantiated via LLM-as-a-judge or other learned evaluators that assess whether the generated output satisfies the criterion-specific requirements.
>
> We construct the current CC-Bench as a clean testbed because it allows precise control of criterion-invariant vs. criterion-sensitive samples. We view this as a first step, and believe CC-ICL offers a general evaluation paradigm for criterion-conditioned reasoning, applicable well beyond the current benchmark.

---

> > ### Author Rebuttal · Reviewer_scc8 · 2026-04-03
> >
> > I thank the authors for their detailed rebuttal. The additional cross-domain evidence and the three-criteria experiment effectively address my concerns regarding the generalization of MCT beyond simple binary memorization.
> >
> > While the provided results demonstrate the effectiveness of the proposed strategy, I remain somewhat concerned about the heavy reliance on fine-tuning to evoke this capability, as well as the inherent difficulty in generalizing from static to dynamic modalities (e.g., the Medical-to-Video gap). However, as the paper successfully formalizes a critical new setting and provides a robust benchmark, I believe its contributions are significant. I will maintain my Weak Accept recommendation.

---

> > > ### Author Response · Authors · 2026-04-06
> > >
> > > We thank Reviewer scc8 for the thoughtful acknowledgement and for maintaining a positive assessment of our CC-ICL setting and CC-Bench contributions.
> > >
> > > Regarding the remaining concern on fine-tuning reliance, we would like to briefly share our perspective. The way ICL capabilities are elicited is closely tied to the base model's inherent ability boundary. For instance, in VL-ICL Bench[1], some models achieve task induction gains simply by increasing the number of shots, while in True-MICL[2] and SymDPO[3], SFT or RL is required to unlock the corresponding ICL capabilities. For criterion adaptation, a capability that current VLMs fundamentally lack, our exploration of training-free alternatives (shot scaling, text prompting) found them ineffective (Sec. 5.4). MCT represents a simple yet effective first step to bridge this gap.
> > >
> > > We agree that achieving training-free, zero-shot criterion adaptation across broader tasks remains an open and important challenge, and we will pursue this as future work. We hope that our formalization of CC-ICL and the diagnostic findings from CC-Bench can draw VLM developers' attention to this critical capability gap in real-world deployment, motivating further research from the community.
> > >
> > > ---
> > > ### **References**
> > > [1] VL-ICL Bench: The Devil in the Details of Multimodal In-Context Learning
> > >
> > > [2] True Multimodal In-Context Learning Needs Attention to the Visual Context
> > >
> > > [3] SymDPO: Boosting In-Context Learning of Large Multimodal Models with Symbol Demonstration Direct Preference Optimization

---

### Decision · Program_Chairs · 2026-04-30

**Decision:**

Accept (regular)

**Comment:**

The paper introduces a new multimodal in-context learning setting, termed Criterion-Conditional In-Context Learning (CC-ICL). Rather than asking whether a model can learn a task from a few-shot context, the paper studies whether the model can dynamically adjust its decision boundary according to the implicit criterion in context while the underlying task semantics remain fixed. This problem formulation is novel and practically meaningful, rather than simply proposing another benchmark. In addition, the proposed CI and CS metrics disentangle robustness and adaptivity, making the evaluation more informative than accuracy alone.

However, the main limitation of the paper is that the current evaluation is mostly focused on binary settings. This is acceptable as an initial attempt to study a new problem setting. The authors also provided strong evidence in the rebuttal to address the reviewers’ concerns, and I encourage them to incorporate these clarifications and additional evidence into the camera-ready version. Given the above considerations, I recommend acceptance.